# Adversarial Examples Exist in Two-Layer ReLU Networks for Low Dimensional Linear Subspaces

Odelia Melamed [*]        Gilad Yehudai[†]        Gal Vardi [‡]

## Abstract

Despite a great deal of research, it is still not well-understood why trained neural networks are highly vulnerable to adversarial examples. In this work we focus on two-layer neural networks trained using data which lie on a low dimensional linear subspace. We show that standard gradient methods lead to non-robust neural networks, namely, networks which have large gradients in directions orthogonal to the data subspace, and are susceptible to small adversarial $L_2$-perturbations in these directions. Moreover, we show that decreasing the initialization scale of the training algorithm, or adding $L_2$ regularization, can make the trained network more robust to adversarial perturbations orthogonal to the data.

## 1 Introduction

Neural networks are observed to be susceptible to adversarial perturbations [Szegedy et al., 2013], often imperceptible by humans. Many works have shown attacks, where adding a very small perturbation to the input may change the prediction of the network [Carlini and Wagner, 2017, Papernot et al., 2017, Athalye et al., 2018]. Other works have shown defense mechanisms called *adversarial training* [Papernot et al., 2016, Madry et al., 2017, Wong and Kolter, 2018]. Despite a great deal of research, it is still not well-understood why neural-network training methods tend towards such *non-robust* solutions.

Several recent works have given theoretical explanation for the existence of adversarial perturbations under different settings. One line of work [Daniely and Shacham, 2020, Bubeck et al., 2021a, Bartlett et al., 2021, Montanari and Wu, 2022] have shown that random networks are susceptible to adversarial perturbations. These results might explain why neural networks are non-robust at initialization, but they do not explain why trained neural networks are non-robust. Recently, Vardi et al. [2022] showed that for data which is *nearly orthogonal*, after training for infinitely many iterations, the implicit bias of neural networks towards margin maximization leads to non-robust solutions. Despite these works, it is still unclear why trained neural networks tend to be non-robust to adversarial perturbations, and specifically what are the assumptions on the input data which leads to non-robustness.

One common belief about "real-life" datasets, is that they approximately lie on a low dimensional "data-manifold" in a high dimensional space. In this setting, the existence of perturbations orthogonal to the data-manifold that change the network's predictions is especially undesired, since such perturbations do not make the input closer to data points from other classes. Indeed, such a perturbation only increases the distance between the input and all "real-life" examples. Shamir et al. [2021] have demonstrated empirically that under such a data-manifold assumption, the decision boundary of a trained classifier clings to the data-manifold in a way that even very small perturbations orthogonal to the manifold can change the prediction.

---

[*]Weizmann Institute of Science, Israel, `odelia.melamed@weizmann.ac.il`

[†]Weizmann Institute of Science, Israel, `gilad.yehudai@weizmann.ac.il`

[‡]TTI-Chicago and the Hebrew University of Jerusalem, `galvardi@ttic.edu`.

37th Conference on Neural Information Processing Systems (NeurIPS 2023).

In this paper we focus on data which lies on a low dimensional "data-manifold". Specifically, we assume that the data lies on a linear subspace $P \subseteq \mathbb{R}^d$ of dimension $d - \ell$ for some $\ell > 0$. We study adversarial perturbations in the direction of $P^\perp$, i.e. orthogonal to the data subspace. We show that the gradient projected on $P^\perp$ is large, and in addition there exist a *universal adversarial perturbation* in a direction orthogonal to $P$. Namely, the same small adversarial perturbation applies to many inputs. The norm of the gradient depends on the term $\frac{\ell}{d}$, while the perturbation size depends on the term $\frac{d}{\ell}$, i.e. a low dimensional subspace implies reduced adversarial robustness. Finally, we also study how changing the initialization scale or adding $L_2$ regularization affects robustness. We show that in our setting, decreasing the initialization scale, or adding a sufficiently large regularization term, can make the network significantly more robust. We also demonstrate empirically the effects of the initialization scale and regularization on the decision boundary. Our experiments suggest that these effects might extend to deeper networks.

## 2   Related Works

Despite extensive research, the reasons for the abundance of adversarial examples in trained neural networks are still not well understood [Goodfellow et al., 2014b, Fawzi et al., 2018, Shafahi et al., 2018, Schmidt et al., 2018, Khoury and Hadfield-Menell, 2018, Bubeck et al., 2019, Allen-Zhu and Li, 2020, Wang et al., 2020, Shah et al., 2020, Shamir et al., 2021, Ge et al., 2021, Wang et al., 2022, Dohmatob and Bietti, 2022]. Below we discuss several prior works on this question.

In a string of works, it was shown that small adversarial perturbations can be found for any fixed input in certain ReLU networks with random weights (drawn from the Gaussian distribution). Building on Shamir et al. [2019], it was shown in Daniely and Shacham [2020] that small adversarial $L_2$-perturbations can be found in random ReLU networks where each layer has vanishing width relative to the previous layer. Bubeck et al. [2021a] extended this result to two-layer neural networks without the vanishing width assumption, and Bartlett et al. [2021] extended it to a large family of ReLU networks of constant depth. Finally, Montanari and Wu [2022] provided a similar result, but with weaker assumptions on the network width and activation functions. These works aim to explain the abundance of adversarial examples in neural networks, since they imply that adversarial examples are common in random networks, and in particular in random initializations of gradient-based methods. However, trained networks are clearly not random, and properties that hold in random networks may not hold in trained networks. Our results also involve an analysis of the random initialization, but we consider the projection of the weights onto the linear subspace orthogonal to the data, and study its implications on the perturbation size required for flipping the output's sign in trained networks.

In Bubeck et al. [2021b] and Bubeck and Sellke [2021], the authors proved under certain assumptions, that overparameterization is necessary if one wants to interpolate training data using a neural network with a small Lipschitz constant. Namely, neural networks with a small number of parameters are not expressive enough to interpolate the training data while having a small Lipschitz constant. These results suggest that overparameterization might be necessary for robustness.

Vardi et al. [2022] considered a setting where the training dataset $\mathcal{S}$ consists of nearly-orthogonal points, and proved that every network to which gradient flow might converge is non-robust w.r.t. $\mathcal{S}$. Namely, building on known properties of the implicit bias of gradient flow when training two-layer ReLU networks w.r.t. the logistic loss, they proved that for every two-layer ReLU network to which gradient flow might converge as the time $t$ tends to infinity, and every point $\mathbf{x}_i$ from $\mathcal{S}$, it is possible to flip the output's sign with a small perturbation. We note that in Vardi et al. [2022] there is a strict limit on the number of training samples and their correlations, as well as the training duration. Here, we have no assumptions on the number of data points and their structure, besides lying on a low-dimensional subspace. Also, in Vardi et al. [2022] the adversarial perturbations are shown to exist only for samples in the training set, while here we show existence of adversarial perturbation for any sample which lies on the low-dimensional manifold.

It is widely common to assume that "real-life data" (such as images, videos, text, etc.) lie roughly within some underlying low-dimensional data manifold. This common belief started many successful research fields such as GAN [Goodfellow et al., 2014a], VAE [Kingma and Welling, 2013], and diffusion [Sohl-Dickstein et al., 2015]. In Fawzi et al. [2018] the authors consider a setting where the high dimensional input data is generated from a low-dimensional latent space. They theoretically analyze the existence of adversarial perturbations on the manifold generated from the latent space,

although they do not bound the norm of these perturbations. Previous works analyzed adversarial perturbations orthogonal to the data manifold. For example, Khoury and Hadfield-Menell [2018] considering several geometrical properties of adversarial perturbation and adversarial training for low dimensional data manifolds. Tanay and Griffin [2016] analyzed theoretically such perturbations for linear networks, and Stutz et al. [2019] gave an empirical analysis for non-linear models. Moreover, several experimental defence methods against adversarial examples were obtained, using projection of it onto the data manifold to eliminate the component orthogonal to the data (see, e.g., Jalal et al. [2017], Meng and Chen [2017], Samangouei et al. [2018]).

Shamir et al. [2021] showed empirically on both synthetic and realistic datasets that the decision boundary of classifiers clings onto the data manifold, causing very close off-manifold adversarial examples. Our paper continues this direction, and provides theoretical guarantees for off-manifold perturbations on trained two-layer ReLU networks, in the special case where the manifold is a linear subspace.

## 3 Setting

**Notations.** We denote by $\mathcal{U}(A)$ the uniform distribution over a set $A$. The multivariate normal distribution with mean $\mu$ and covariance $\Sigma$ is denoted by $\mathcal{N}(\mu, \Sigma)$, and the univariate normal distribution with mean $a$ and variance $\sigma^2$ is denoted by $\mathcal{N}(a, \sigma^2)$. The set of integers $\{1, .., m\}$ is denoted by $[m]$. For a vector $v \in \mathbb{R}^n$, we define $v_{i:i+j}$ to be the $j + 1$ coordinates of $v$ starting from $i$ and ending with $i + j$. For a vector $x$ and a linear subspace $P$ we denote by $P^\perp$ the subspace orthogonal to $P$, and by $\Pi_{P^\perp}(x)$ the projection of $x$ on $P^\perp$. We denote by $\mathbf{0}$ the zero vector. We use $I_d$ for the identity matrix of size $d$.

### 3.1 Architecture and Training

In this paper we consider a two-layer fully-connected neural network $N : \mathbb{R}^d \to \mathbb{R}$ with ReLU activation, input dimension $d$ and hidden dimension $m$:

$$N(x, \mathbf{w}_{1:m}) = \sum_{i=1}^{m} u_i \sigma(w_i^\top x) .$$

Here, $\sigma(z) = \max(z, 0)$ is the ReLU function and $\mathbf{w}_{1:m} = (w_1, \ldots, w_m)$. When $\mathbf{w}_{1:m}$ is clear from the context, we will write for short $N(x)$.

We initialize the first layer using standard Kaiming initialization [He et al., 2015], i.e. $w_i \sim \mathcal{N}\left(\mathbf{0}, \frac{1}{d} I_d\right)$, and the output layer as $u_i \sim \mathcal{U}\left(\left\{-\frac{1}{\sqrt{m}}, \frac{1}{\sqrt{m}}\right\}\right)$ for every $i \in [m]$. Note that in standard initialization, each $u_i$ would be initialized normally with a standard deviation of $\frac{1}{\sqrt{m}}$, for ease of analysis we fix the initial value to be equal to the standard deviation where only the sign is random.

We consider a dataset with binary labels. Given a training dataset $(x_1, y_1), \ldots, (x_r, y_r) \in \mathbb{R}^d \times \{-1, 1\}$ we train w.r.t. the logistic loss (a.k.a. binary cross entropy): $L(q) = \log(1 + e^{-q})$, and minimize the empirical error:

$$\min_{w_1, \ldots, w_m} \sum_{i=1}^{r} L\left(y_i \cdot N(x_i, \mathbf{w}_{1:m})\right) .$$

We assume throughout the paper that the network is trained using either gradient descent (GD) or stochastic gradient descent (SGD). Our results hold for both training methods. We assume that only the weights of the first layer (i.e. the $w_i$'s) are trained, while the weights of the second layer (i.e. the $u_i$'s) are fixed.

### 3.2 Assumptions on the Data

Our main assumption in this paper is that the input data lie on a low dimensional manifold, which is embedded in a high dimensional space. Specifically, we assume that this "data manifold" is a linear subspace, denoted by $P$, which has dimension $d - \ell$. We denote by $\ell$ the dimension of the

data "off-manifold", i.e. the linear subspace orthogonal to the data subspace, which is denoted by $P^\perp$. In this work we study adversarial perturbations in $P^\perp$. Note that adding a perturbation from $P^\perp$ of any size to an input data point which changes its label is an unwanted phenomenon, because this perturbation is orthogonal to any possible data point from both possible labels. We will later show that under certain assumptions there exists an adversarial perturbation in the direction of $P^\perp$ which also has a small norm. This reason for this assumption is so that the projection of the first layer weights on $P^\perp$ remain fixed during training. An interesting question is to consider general "data manifolds", which we elaborate on in Section 7.

To demonstrate that the low-dimensional data assumption arises in practical settings, in Figure 1 we plot the cumulative variance of the MNIST and CIFAR10 datasets, projected on a linear manifold. These are calculated by performing PCA on the entire datasets, and summing over the square of the singular values from largest to smallest. For CIFAR10, the accumulated variance reaches $90\%$ at 98 components, and $95\%$ at 216 components. For MNIST, the accumulated variance reaches $90\%$ at 86 components, and $95\%$ at 153 components. This indicates that both datasets can be projected to a much smaller linear subspace, without losing much of the information.

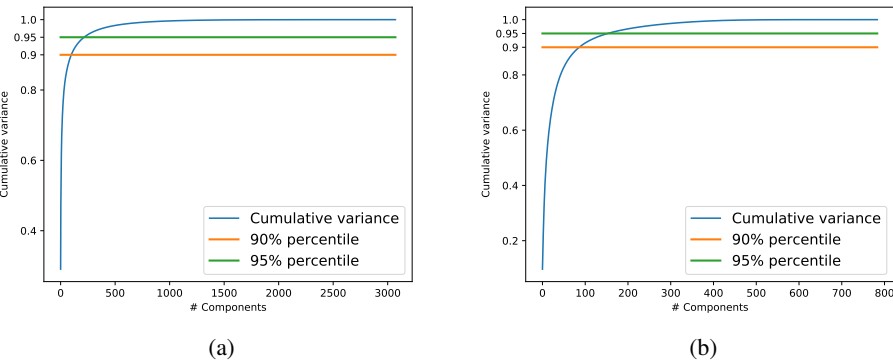

(a)                                (b)

Figure 1: The cumulative variance for the (a) CIFAR10; and (b) MNIST datasets, calculated by performing PCA on the entire datasets, and summing over the square of the singular values from largest to smallest.

**Remark 3.1** (On the Margin of the Network). *Given a neural network $N : \mathbb{R}^d \to \mathbb{R}$ and a dataset $(x_1, y_1), \ldots, (x_r, y_r)$ with binary labels which the network label correctly, we define the margin of the network as $\gamma := \min_{i \in [r]} y_i N(x_i)$.*

*In our setting, it is possible to roughly estimate the margin without assuming much about the data, besides its boundedness. Note that the gradient of the loss decays exponentially with the output of the network, because $\left| \frac{\partial L(q)}{\partial q} \right| = \left| \frac{-qe^{-q}}{1+e^{-q}} \right|$. Hence, if we train for at most polynomially many iterations and label all the data points correctly (i.e. the margin is larger than $0$), then training effectively stops after the margin reaches $O(\log^2(d))$. This is because if the margin is $\log^2(d)$, then the gradient is of size:*

$$\left| L'(\log^2(d)) \right| = \left| \frac{-\log^2(d) e^{-\log^2(d)}}{1 + e^{-\log^2(d)}} \right| \le \log^2(d) \cdot d^{-\log(d)} \, ,$$

*which is smaller than any polynomials in $d$. This means that all the data points on the margin (which consists of at least one point) will have an output of $O(\mathrm{polylog}(d))$.*

*The number of points which lie exactly on the margin is difficult to assess, since it may depend on both the dataset and the model. Some empirical results in this direction are given in Vardi et al. [2022], where it is observed (empirically) that for data sampled uniformly from a sphere and trained with a two-layer network, over $90\%$ of the input samples lie on the margin. Also, in Haim et al. [2022] it is shown that for CIFAR10, a large portion of the dataset lies on the margin.*

# 4 Large Gradient Orthogonal to the Data Subspace

One proxy for showing non-robustness of models, is to show that their gradient w.r.t. the input data is large (cf. Bubeck et al. [2021b], Bubeck and Sellke [2021]). Although a large gradient does not guarantee that there is also an adversarial perturbation, it is an indication that a small change in the input might significantly change the output. Moreover, by assuming smoothness of the model, it is possible to show that having a large gradient may suffice for having an adversarial perturbation.

In this section we show that training a network on a dataset which lies entirely on a linear subspace yields a large gradient in a direction which is orthogonal to this subspace. Moreover, the size of the gradient depends on the dimension of this subspace. Specifically, the smaller the dimension of the data subspace, the larger the gradient is in the orthogonal direction. Our main result in this section is the following:

**Theorem 4.1.** *Suppose that a network $N(x) = \sum_{i=1}^{m} u_i \sigma(\langle w_i, x \rangle)$ is trained on a dataset which lies on a linear subspace $P \subset \mathbb{R}^d$ of dimension $d - \ell$ for $\ell \geq 1$, and let $x_0 \in P$. Let $S = \{i \in [m] : \langle w_i, x_0 \rangle \geq 0\}$, and let $k := |S|$. Then, w.p $\geq 1 - e^{-\ell/16}$ (over the initialization) we have:*

$$\left\| \Pi_{P^\perp} \left( \frac{\partial N(x_0)}{\partial x} \right) \right\| \geq \sqrt{\frac{k\ell}{2md}} \, .$$

The full proof can be found in Appendix B. Here we provide a short proof intuition: First, we use a symmetry argument to show that it suffices to consider w.l.o.g. the subspace $M := \text{span}\{e_1, \ldots, e_{d-\ell}\}$, where $e_i$ are the standard unit vectors. Next, we note that since the dataset lies on $M$, only the first $d - \ell$ coordinates of each weight vector $w_i$ are trained, while the other $\ell$ coordinates are fixed at their initial value. Finally, using standard concentration result on Gaussian random variables we can lower bound the norm of the gradient. Note that our result shows that there might be a large gradient orthogonal to the data subspace. This correspond to "off-manifold" adversarial examples, while the full gradient (i.e. without projecting on $P^\perp$) might be even larger.

The lower bound on the gradient depends on two terms: $\frac{k}{m}$ and $\frac{\ell}{d}$. The first term is the fraction of active neurons for the input $x_0$, i.e. the neurons whose inputs are non-negative. Note that inactive neurons do not increase the gradient, since they do not affect the output. The second term corresponds to the fraction of directions orthogonal to the data. The larger the dimension of the orthogonal subspace, the more directions in which it is possible to perturb the input while still being orthogonal to the data. If both of these terms are constant, i.e. there is a constant fraction of active neurons, and "off-manifold" directions, we can give a more concrete bound on the gradient:

**Corollary 4.1.** *For $\ell = \Omega(d)$, $k = \Omega(m)$, in the setting of Theorem 4.1, with probability $\geq 1 - e^{-\Omega(d)}$ we have:*

$$\left\| \Pi_{P^\perp} \left( \frac{\partial N(x_0)}{\partial x} \right) \right\| = \Omega(1) \, .$$

Consider the case where the norm of each data point is $\Theta(\sqrt{d}) = \Theta(\sqrt{\ell})$, i.e. every coordinate is of size $\Theta(1)$. By Remark 3.1, for a point $x_0$ on the margin, its output is of size $\text{polylog}(d)$. Therefore, for the point $x_0$, gradient of size $\Omega(1)$ corresponds to an adversarial perturbation of size $\text{polylog}(d)$, which is much smaller than $\|x_0\| = \Theta(\sqrt{d})$. We note that this is a rough and informal estimation, since, as we already discussed, a large gradient at $x_0$ does not necessarily imply that an adversarial perturbation exists. In the next section, we will prove the existence of adversarial perturbations.

# 5 Existence of an Adversarial Perturbation

In the previous section we have shown that at any point $x_0$ which lies on the linear subspace of the data $P$, there is a large gradient in the direction of $P^\perp$. In this section we show that not only the gradient is large, there also exists an adversarial perturbation in the direction of $P^\perp$ which changes the label of a data point from $P$ (under certain assumptions). The main theorem of this section is the following:

**Theorem 5.1.** *Suppose that a network $N(x) = \sum_{i=1}^{m} u_i \sigma(\langle w_i, x \rangle)$ is trained on a dataset which lies on a linear subspace $P \subseteq \mathbb{R}^d$ of dimension $d - \ell$, where $\ell \geq 32(m-1)\log(m^2 d)$. Let $x_0 \in P$,*

*and denote $y_0 := sign(N(x_0))$. Let $I_- := \{i \in [m] : u_i < 0\}$ and $I_+ := \{i \in [m] : u_i > 0\}$, and denote $k_- := |\{i \in I_- : \langle w_i, x_0 \rangle \geq 0\}|$, and $k_+ := |\{i \in I_+ : \langle w_i, x_0 \rangle \geq 0\}|$. Let $k_{y_0} = k_-$ if $y_0 = 1$ and $k_{y_0} = k_+$ if $y_0 = -1$. For $w \in \mathbb{R}^d$ denote $\hat{w} := \Pi_{P^\perp}(w)$, and denote the perturbation*

$$z := y_0 \cdot \alpha \left( \sum_{i \in I_-} \hat{w}_i - \sum_{i \in I_+} \hat{w}_i \right) \text{ where } \alpha = \frac{8\sqrt{m}dN(x_0)}{\ell k_{y_0}}. \text{ Then, w.p. } \geq 1 - 5(me^{-\ell/16} + d^{-1/2})$$

*we have that $\|z\| \leq 8\sqrt{2}N(x_0) \cdot \frac{m}{k_{y_0}} \cdot \sqrt{\frac{d}{\ell}}$ and:*

$$sign(N(x_0 + z)) \neq sign(N(x_0)) \,.$$

The full proof can be found in Appendix C. Here we give a short proof intuition: As in the previous section, we show using a symmetry argument that w.l.o.g. we can assume that $P = \text{span}\{e_1, \ldots, e_{d-\ell}\}$.

Now, given the perturbation $z$ from Theorem 5.1 we want to understand how adding it to the input changes the output. Suppose that $y_0 = 1$. We can write

$$N(x_0 + z) = \sum_{i \in I_-} u_i \sigma(\langle w_i, x_0 \rangle + \langle w_i, z \rangle) + \sum_{i \in I_+} u_i \sigma(\langle w_i, x_0 \rangle + \langle w_i, z \rangle)$$

We can see that for all $i$:

$$\langle w_i, z \rangle = \alpha \cdot \langle w_i, \sum_{j \in I_-} \Pi_{P^\perp}(w_j) - \sum_{j \in I_+} \Pi_{P^\perp}(w_j) \rangle$$

$$= -\alpha \cdot \langle w_i, \sum_{j=1}^{m} \text{sign}(u_j)\Pi_{P^\perp}(w_j) \rangle.$$

For $i \in I_-$ we can write:

$$\langle w_i, z \rangle = \alpha \left\| \Pi_{P^\perp}(w_i) \right\|^2 - \alpha \langle \Pi_{P^\perp}(w_i), \sum_{j \neq i} \text{sign}(u_j)\Pi_{P^\perp}(w_j) \rangle \,,$$

and using a similar calculation, for $i \in I_+$ we can write:

$$\langle w_i, z \rangle = -\alpha \left\| \Pi_{P^\perp}(w_i) \right\|^2 - \alpha \langle \Pi_{P^\perp}(w_i), \sum_{j \neq i} \text{sign}(u_j)\Pi_{P^\perp}(w_j) \rangle \,.$$

Using concentration inequalities of Gaussian random variables, and the fact that $\Pi_{P^\perp}(w_i)$ did not change from their initial values, we can show that:

$$\left| \langle \Pi_{P^\perp}(w_i), \sum_{j \neq i} \text{sign}(u_j)\Pi_{P^\perp}(w_j) \rangle \right| \approx \frac{\sqrt{\ell m}}{d} \,,$$

while $\|\Pi_{P^\perp}(w_i)\|^2 \approx \frac{\ell}{d}$. Thus, for a large enough value of $\ell$ we have that $\langle w_i, z \rangle \leq 0$ for $i \in I_+$ and $\langle w_i, z \rangle \approx \alpha \cdot \frac{\sqrt{\ell}}{d} \cdot (\sqrt{\ell} - \sqrt{m})$ for $i \in I_-$.

From the above calculations we can see that adding the perturbation $z$ does not increase the output of the neurons with a positive second layer. On the other hand, adding $z$ can only increase the input of the neurons with negative second layer, and for those neurons which are also active it increases their output as well if we assume that $\ell > m$. This means, that if there are enough active neurons with a negative second layer (denoted by $k_-$ in the theorem), then the perturbation can significantly change the output. In the proof we rely only on the active negative neurons to change the label of the output (for the case of $y_0 = 1$, if $y_0 = -1$ we rely on the active positive neurons). Note that the active positive neurons may become inactive, and the inactive negative neurons may become active. Without further assumptions it is not clear what is the size of the perturbation to make this change for every neuron. Thus, the only neurons that are guaranteed to help change the label are the active negative ones, which by our assumptions on $\ell$ are guaranteed to increase their output.

Note that our perturbation is *not* in the direction of the gradient w.r.t. the input. The direction of the gradient would be the sum of all the active neurons, i.e. the sum (with appropriate signs) over all $i \in [m]$ such that $\langle w_i, x_0 \rangle \geq 0$. Our analysis would not have worked with such a perturbation, because we could not guarantee that inactive neurons would stay inactive.

The assumption that $\ell = \Omega(M)$ (up to log factors) is a technical limitation of our proof technique. We note that such an assumption is also used in other theoretical papers about adversarial perturbations (e.g. Daniely and Shacham [2020]).

Note that the direction of the perturbation $z$ does not depend on the input data $x_0$, only its size depends on $x_0$. In fact, Theorem 5.1 shows that there is a single universal direction for an adversarial perturbation that can flip the label of any data point in $P$. The size of the perturbation depends on the dimension of the linear subspace of the data, the number of active neurons for $x_0$, the total number of neurons in the network and the size of the output. In the following corollary we give a specific bound on the size of the perturbations under assumptions on the different parameters of the problem:

**Corollary 5.1.** *In the setting of Theorem 5.1, assume in addition that $\ell = \Theta(d)$ and $k_{y_0} = \Theta(m)$. Then, there exists a perturbation $z$ such that w.p. $\geq 1 - 5\left(me^{-\ell/16} + d^{-1/2}\right)$ we have $\|z\| = O(N(x_0))$ and:*
$$sign(N(x_0 + z)) \neq sign(N(x_0)) .$$

The above corollary follows directly by noticing from Theorem 5.1 that:
$$\|z\| \leq O\left(N(x_0) \cdot \frac{m}{k_{y_0}} \cdot \sqrt{\frac{d}{\ell}}\right) = O(N(x_0)) ,$$

where we plugged in the additional assumptions. The assumptions in the corollary above are similar to the assumptions in Corollary 4.1. Namely, that the dimension of the data subspace is a constant fraction from the dimension of the entire space, and the number of active neurons is a constant fraction of the total number of neurons. Note that here we only consider active neurons with a specific sign in the second layer.

Note that the size of the perturbation in Corollary 5.1 is bounded by $N(x_0)$. By Remark 3.1, the output of the network for data points on the margin can be at most $O(\log^2(d))$, since otherwise the network would have essentially stopped training. Therefore, if we consider an input $x_0$ on the margin, and $\|x_0\| = \Theta(\sqrt{d}) = \Theta(\sqrt{\ell})$, then the size of the adversarial perturbation is much smaller than $\|x_0\|$. For any other point, without assuming it is on the margin, and since we do not assume anything about the training data (except for being in $P$), we must assume that the size of the perturbation required to change the label will depend on the size of the output.

## 6 The Effects of the Initialization Scale and Regularization on Robustness

In Section 5, we presented the inherent vulnerability of trained models to small perturbations in a direction orthogonal to the data subspace. In this section, we return to a common proxy for robustness that we considered in Section 4 – the gradient at an input point $x_0$. We suggest two ways that might improve the robustness of the model in the direction orthogonal to the data, by decreasing an upper bound of the gradient in this direction. We first upper bound the gradient of the model in the general case where we initialize $w_i \sim \mathcal{N}(\mathbf{0}, \beta^2 I_d)$, and later discuss strategies to use this upper bound for improving robustness.

**Theorem 6.1.** *Suppose that a network $N(x) = \sum\limits_{i=1}^{m} u_i\sigma(\langle w_i, x\rangle)$ is trained on a dataset which lies on a linear subspace $P \subseteq \mathbb{R}^d$ of dimension $d - \ell$ for $\ell \geq 1$, and assume that the weights $w_i$ are initialized from $\mathcal{N}(\mathbf{0}, \beta^2 I_d)$. Let $x_0 \in P$, let $S = \{i \in [m] : \langle w_i, x_0\rangle \geq 0\}$, and let $k = |S|$. Then, w.p. $\geq 1 - e^{-\frac{\ell}{16}}$ we have:*
$$\left\|\Pi_{P^\perp}\left(\frac{\partial N(x_0)}{\partial x}\right)\right\| \leq \beta\sqrt{\frac{2k\ell}{m}} .$$

The full proof uses the same concentration bounds ideas as the lower bound proof and can be found in Appendix D. This bound is a result of the untrained weights: since the projection of the data points on $P^\perp$ is zero, the projection of the weights vectors on $P^\perp$ are not trained and are fixed at their initialization. We note that Theorem 4.1 readily extends to the case of initialization from $\mathcal{N}(\mathbf{0}, \beta^2 I_d)$, in which case the lower bound it provides matches the upper bound from Theorem 6.1 up to a constant factor. In what follows, we suggest two ways to affect the post-training weights in the $P^\perp$ direction: (1) To initialize the weights vector using a smaller-variance initialization, and (2) Add an $L_2$-norm regularization on the weights. We next analyze their effect on the upper bound.

## 6.1 Small Initialization Variance

From Theorem 6.1, one can conclude a strong result about the model's gradient without the dependency of its norm on $\ell$ and $k$.

**Corollary 6.1.** *For $\beta = \frac{1}{d\sqrt{2}}$, in the settings of Theorem 6.1, with probability $\geq 1 - e^{-\frac{\ell}{16}}$ we have*

$$\left\| \Pi_{P^\perp} \left( \frac{\partial N(x_0)}{\partial x} \right) \right\| \leq \frac{1}{\sqrt{d}} .$$

The proof follows directly from Theorem 6.1, by noticing that $k \leq m$ and $\ell \leq d$. Consider for example an input $x_0 \in P$ with $\|x_0\| = \Theta(\sqrt{d})$, and suppose that $N(x_0) = \Omega(1)$. The above corollary implies that if the initialization has a variance of $1/d^2$ (rather than the standard choice of $1/d$) then the gradient is of size $O\left(1/\sqrt{d}\right)$. Thus, it corresponds to perturbations of size $\Omega(\sqrt{d})$, which is the same order as $\|x_0\|$.

## 6.2 $L_2$ Regularization

We consider another way to influence the projection onto $P^\perp$ of the trained weights vectors: adding $L_2$ regularization while training. We will update the logistic loss function by adding an additive factor $\frac{1}{2}\lambda \|\mathbf{w}_{1:m}\|^2$. For a dataset $(x_1, y_1), .., (x_r, y_r)$, we now train over the following objective:

$$\sum_{j=1}^{r} L(y_j \cdot N(x_j, \mathbf{w}_{1:m}))) + \frac{1}{2}\lambda \|\mathbf{w}_{1:m}\|^2 .$$

This regularization will cause the previously untrained weights to decrease in each training step which will decrease the upper bound on the projection of the gradient:

**Theorem 6.2.** *Suppose that a network $N(x) = \sum_{i=1}^{m} u_i \sigma(\langle w_i, x \rangle)$ is trained for $T$ training steps, using $L_2$ regularization with parameter $\lambda \geq 0$ and step size $\eta > 0$, on a dataset which lies on a linear subspace $P \subseteq \mathbb{R}^d$ of dimension $d - \ell$ for $\ell \geq 1$, starting from standard initialization (i.e., $w_i \sim \mathcal{N}(\mathbf{0}, \frac{1}{d}I_d)$). Let $x_0 \in P$, let $S = \{i \in [m] : \langle w_i, x_0 \rangle \geq 0\}$, and let $k := |S|$. Then, w.p. $\geq 1 - e^{-\frac{\ell}{16}}$ we have*

$$\left\| \Pi_{P^\perp} \left( \frac{\partial N(x_0)}{\partial x} \right) \right\| \leq (1 - \eta\lambda)^T \sqrt{\frac{2k\ell}{md}} .$$

The full proof can be found in Appendix D.1. The main idea of the proof is to observe the projection of the weights on $P^\perp$ changing during training. As before, we assume w.l.o.g. that $P = \text{span}\{e_1, \ldots, e_{d-\ell}\}$ and denote by $\hat{w}_i := \Pi_{P^\perp}(w_i)$. During training, the weight vector's last $\ell$ coordinates are only affected by the regularization term of the loss. These weights decrease in a constant multiplicand of the previous weights. Thus, we can conclude that for every $t \geq 0$ we have: $\hat{w}_i^{(t)} = (1 - \eta\lambda)^t \hat{w}_i^{(0)}$, where $\hat{w}_i^{(t)}$ is the $i$-th weight vector at time $t$. It implies that our setting is equivalent to initializing the weights with standard deviation $\frac{(1-\eta\lambda)^T}{\sqrt{d}}$ and training the model without regularization for $T$ steps. As a result, we get the following corollary:

**Corollary 6.2.** *For $(1 - \eta\lambda)^T \leq \frac{1}{\sqrt{2d}}$, in the settings of 6.2, w.p. $\geq 1 - e^{-\frac{\ell}{16}}$ we get that:*

$$\left\| \Pi_{P^\perp} \left( \frac{\partial N(x_0)}{\partial x} \right) \right\| \leq \frac{1}{\sqrt{d}} .$$

## 6.3 Experiments

In this section, we present our robustness-improving experiments. [4] We explore our methods on two datasets: (1) A 7-point dataset on a one-dimensional linear subspace in a two-dimensional input

---

[4]For the code of the experiments see https://github.com/odeliamel/off-manifold-robustness

space, and (2) A 25-point dataset on a two-dimensional linear subspace in a three-dimensional input space. In Figures 2 and 3 we present the boundary of a two-layer ReLU network trained over these two datasets. We train the networks until reaching a constant positive margin. We note that unlike our theoretical analysis, in the experiments in Figure 2 we trained all layers and initialize the weights using the default PyTorch initialization, to verify that the observed phenomena occur also in this setting. In the experiment in Figure 3 we use a different initialization scale for the improving effect to be smaller and visualized easily. In Figures 2a and 3a we trained with default settings. In Figures 2b and 3b we initialized the weights using an initialization with a smaller variance (i.e., initialization divided by a constant factor). Finally, in Figures 2c and 3c we train with $L_2$ regularization.

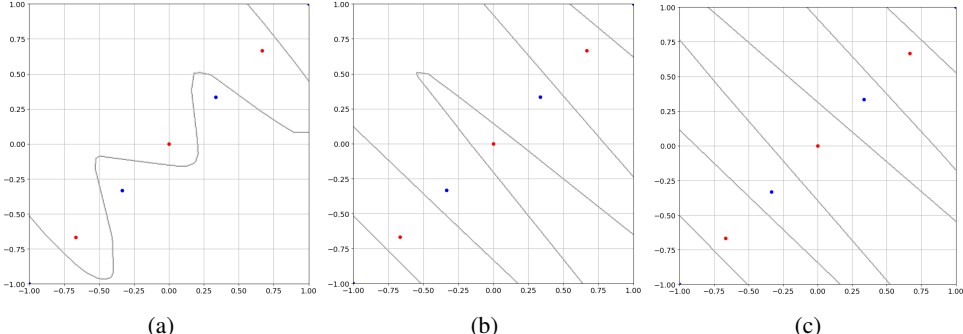

(a)                     (b)                     (c)

Figure 2: **Experiments on a one-dimensional dataset.** We plot the dataset points and the decision boundary in 3 settings: (a) Vanilla trained network, (b) The network's weight are initialized from a smaller variance distribution, and (c) Training with regularization.

Consider the adversarial perturbation in the direction $P^\perp$, orthogonal to the data subspace, in Figures 2 and 3. In figure (a) of each experiment, we can see that a rather small adversarial perturbation is needed to cross the boundary in the subspace orthogonal to the data. In the middle figure (b), we see that the boundary in the orthogonal subspace is much further. This is a direct result of the projection of the weights onto this subspace being much smaller. In the right experiment (c), we can see a similar effect created by regularization. In Appendix E we add the full default-scaled experiment in the two-dimensional setting to demonstrate the robustness effect. There, in both the small-initialization and regularization experiments, the boundary lines are almost orthogonal to the data subspace. In Appendix E we also conduct further experiments with deeper networks and standard PyTorch initialization, showing that our theoretical results are also observed empirically in settings going beyond our theory.

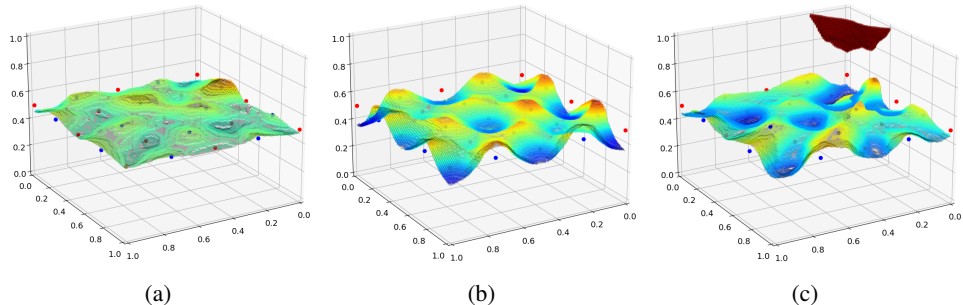

(a)                     (b)                     (c)

Figure 3: **Experiments on two-dimensional dataset demonstrating a smaller robustness effect.** We plot the dataset points and the decision boundary in 3 settings: (a) Vanilla trained network, (b) The network's weights are initialized from a smaller variance distribution, and (c) Training with regularization. Colors are used to emphasise the values in the $z$ axis.

In Figure 4 we plot the distance from the decision boundary for different initialization scales of the first layer. We trained a 3-layer network, initialized using standard initialization except for the first layer which is divided by the factor represented in the $X$-axis. After training, we randomly picked 200 points and used a standard projected gradient descent adversarial attack to change the label of

each point, which is described in the $Y$-axis (perturbation norm, with error bars). The datasets are: (a) Random points from a sphere with 28 dimensions, which lies in a space with 784 dimensions; and (b) MNIST, where the data is projected on 32 dimensions using PCA. The different lines are adversarial attacks projected either on data subspace, on its orthogonal subspace, or without projection. It can be seen that small initialization increases robustness off the data subspace, and also on the non-projected attack, while having almost no effect for the attacks projected on the data subspace.

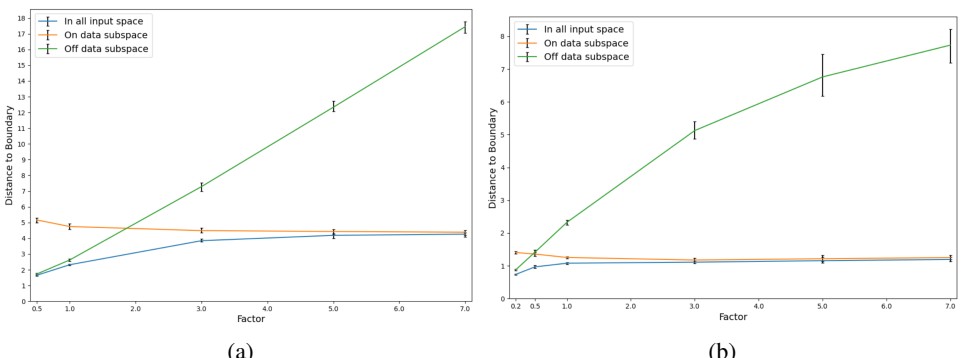

(a)                            (b)

Figure 4: **The distance to the decision boundary for different initializations of the first layer.** The $X$-axis represents the factor which the initialization of the first layer is divided by. The $Y$-axis shows the size of a standard perturbed gradient descent adversarial attack to change the label of each point for 200 randomly picked points. The datasets are: (a) Random points from a sphere with 28 dimensions, which lies in a space with 784 dimensions; and (b) MNIST, where the data is projected on 32 dimensions using PCA. The different lines are adversarial attacks projected either on data subspace, on its orthogonal subspace, or without projection.

## 7   Conclusions and Future Work

In this paper we considered training a two-layer network on a dataset lying on $P \subseteq \mathbb{R}^d$ where $P$ is a $d - \ell$ dimensional subspace. We have shown that the gradient of any point $x_0 \in P$ projected on $P^\perp$ is large, depending on the dimension of $P$ and the fraction of active neurons on $x_0$. We additionally showed that there exists an adversarial perturbation in the direction of $P^\perp$. The size of the perturbation depends in addition on the output of the network on $x_0$, which by Remark 3.1 should be poly-logarithmic in $d$, at least for points which lie on the margin of the network. Finally, we showed that by either decreasing the initialization scale or adding $L_2$ regularization we can make the network robust to "off-manifold" perturbations, by decreasing the gradient in this direction.

One interesting question is whether our results can be generalized to other manifolds, beyond linear subspaces. We state this as an informal open problem:

**Open Problem 7.1.** *Let $M$ be a manifold, and $\mathcal{D}$ a distribution over $M \times \{\pm 1\}$. Suppose we train a network $N : \mathbb{R}^d \to \mathbb{R}$ on a dataset sampled from $\mathcal{D}$. Let $x_0 \in M$, then under what conditions on $M$, $\mathcal{D}$ and $N$, there exists a small adversarial perturbation in the direction of $(T_{x_0} M)^\perp$, i.e. orthogonal to the tangent space $T_{x_0} M$, of $M$ at $x_0$.*

Our result can be seen as a special case of this conjecture, where at all points $x, x' \in M$, the tangent spaces are equal $T_x M = T_{x'} M$. Another future direction would be to analyze deeper networks, or different architectures such as convolutions. Finally, it would also be interesting to analyze robustness of trained networks w.r.t. different norms such as $L_1$ or $L_\infty$.

## Acknowledgments and Disclosure of Funding

We thank Ohad Shamir for the many helpful discussions about this work. We would also like to thank Michal Irani for contributing computational resources. GY was supported in part by the European Research Council (ERC) grant 754705 . GV acknowledges the support of the NSF and the Simons Foundation for the Collaboration on the Theoretical Foundations of Deep Learning.

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

# A   Rotation Invariance w.r.t. the Initialized Weights

In this paper, we analyze neural networks trained on high-dimensional data that lies on a low dimensional linear subspace denoted by $P$. We assume that the dimension of $P$ is $d - \ell$. Throughout the paper it will be more convenient to analyze data which lies on the subspace $M = \text{span}(\{e_1, \ldots, e_{d-\ell}\})$, because then the "off manifold" directions correspond exactly to certain coordinates of the input. In this section we show that we can essentially analyze the data as if it is rotated to lie on $M$, and it would imply the same consequences as the original data from $P$.

**Theorem A.1.** *Let $P \subseteq \mathbb{R}^d$ be a subspace of dimension $d - \ell$, and let $M = \text{span}\{e_1, \ldots, e_{d-\ell}\}$. Let $R$ be an orthogonal matrix such that $R \cdot P = M$, let $X \subseteq P$ be a training dataset and let $X_R = \{R \cdot x : x \in X\}$. Assume we train a neural network $N(x) = \sum_{i=1}^m u_i \sigma(w_i^\top x)$ as explained in Section 3, and denote by $N^X$ and $N^{X_R}$ the network trained on $X$ and $X_R$ respectively for the same number of iterations. Let $x_0 \in P$, then we have:*

1. *W.p. $p$ (over the initialization) we have $\left\| \Pi_{P^\perp} \left( \frac{\partial N^X(x_0)}{\partial x} \right) \right\| \geq c$ (resp. $\leq c$) for some $c \in \mathbb{R}$, iff w.p. $p$ also $\left\| \Pi_{M^\perp} \left( \frac{\partial N^{X_R}(Rx_0)}{\partial x} \right) \right\| \geq c$ (resp. $\leq c$).*

2. *For any $c, p \geq 0$, w.p. $p$ (over the initialization) there exists $z \in P^\perp$ with $\|z\| = c$ such that $\text{sign}(N^X(x_0 + z)) \neq \text{sign}(N^X(x_0))$, iff w.p. $p$ there exists $z' \in M^\perp$ with $\|z'\| = c$ such that $\text{sign}(N^{X_R}(Rx_0 + z')) \neq \text{sign}(N^{X_R}(Rx_0))$.*

*Proof.* Denote by $\mathbf{w}_{1:m} := (w_1, \ldots, w_m)$ and by $R\mathbf{w}_{1:m} = (Rw_1, \ldots, Rw_m)$. Let $\mathbf{w}_{1:m}^{(t)}$ the weights of the network trained on the dataset $X$ where $\mathbf{w}_{1:m}^{(0)}$ is some initialization, and $\tilde{\mathbf{w}}_{1:m}^{(t)} = (\tilde{w}_1^{(t)}, \ldots, \tilde{w}_m^{(t)})$ the weights of the network trained on $X_R$ and initialized at $R\mathbf{w}_{1:m}^{(0)}$. In the proof, when taking derivatives w.r.t. the $w_i$'s we will explicitly write $N(x, \mathbf{w}_{1:m})$.

We first show by induction on the number of training steps that $\tilde{\mathbf{w}}_{1:m}^{(t)} = R\mathbf{w}_{1:m}^{(t)}$. For $t = 0$ it is clear by the assumption on the initialization. Assume it is true for $t$, then we have for some $x \in X$:

$$
\begin{aligned}
\frac{\partial N(Rx, \tilde{\mathbf{w}}_{1:m}^{(t)})}{\partial w_i} &= u_i \sigma'(\langle \tilde{w}_i^{(t)}, Rx \rangle) Rx \\
&= u_i \sigma'(\langle Rw_i^{(t)}, Rx \rangle) Rx \\
&= u_i \sigma'(\langle w_i^{(t)}, x \rangle) Rx \\
&= R \cdot \frac{\partial N(x, \mathbf{w}_{1:m}^{(t)})}{\partial w_i} \ .
\end{aligned}
$$

This is true for every $i \in [m]$ and for every $x \in X$. Also note that by our induction assumption we have:

$$
N(x, \mathbf{w}_{1:m}^{(t)}) = \sum_{i=1}^m u_i \sigma(\langle w_i^{(t)}, x \rangle) = \sum_{i=1}^m u_i \sigma(\langle Rw_i^{(t)}, Rx \rangle) = N(Rx, \tilde{\mathbf{w}}_{1:m}^{(t)}) \ . \tag{1}
$$

Finally, the derivative of the loss on a single data point $x \in X$ with label $y$ can be written as:

$$
\frac{\partial L\left( N(x, \mathbf{w}_{1:m}^{(t)}) \cdot y \right)}{\partial w_i} = L'\left( N(x, \mathbf{w}_{1:m}^{(t)}) \cdot y \right) \cdot \frac{\partial N(x, \mathbf{w}_{1:m}^{(t)})}{\partial w_i} \ ,
$$

where the first term depends only on the value of $N(x, \mathbf{w}_{1:m}^{(t)})$. Hence, taking a single gradient step of $N$ with weights $\mathbf{w}_{1:m}^{(t)}$ and dataset $X$ will change the weights by the same term up to multiplication by $R$ as if taking a gradient step with with weights $\tilde{\mathbf{w}}_{1:m}^{(t)}$ and dataset $X_R$. This finishes the induction.

Let $\mathbf{w}_{1:m}^{(0)}$ be an initialization for the training of $N^X$, where there exists $z \in P^\perp$ with $\|z\| = c$ such that $\text{sign}(N^X(x_0 + z)) \neq \text{sign}(N^X(x_0))$. Then, by Eq. (1) the initialization $R\mathbf{w}_{1:m}^{(0)}$ for the training of $N^{X_R}$ is such that for $z' = Rz$ we have $\|z'\| = c$ and $\text{sign}(N^{X_R}(Rx_0 + z')) \neq \text{sign}(N^{X_R}(Rx_0))$. This argument holds also in the opposite direction. Let $A \subseteq \{\mathbf{w}_{1:m} \in \mathbb{R}^{d \cdot m}\}$ be the set of all

initializations to $N^X$ where there exists $z \in P^\perp$ with $\|z\| = c$ such that $\mathrm{sign}(N^X(x_0 + z)) \neq \mathrm{sign}(N^X(x_0))$, then by the above the set $R \cdot A = \{R\mathbf{w}_{1:m} : \mathbf{w}_{1:m} \in A\}$ are exactly all the initializations to $N^{X_R}$ where there exists $z' \in M^\perp$ with $\|z'\| = c$ such that $\mathrm{sign}(N^{X_R}(Rx_0 + z')) \neq \mathrm{sign}(N^{X_R}(Rx_0))$. Since we initialize the $w_i$'s using a Gaussian initialization which is spherically symmetric, we have that $\Pr(A) = \Pr(RA)$. This proves item (2). Item (1) follows from similar arguments (which we do not repeat for conciseness). □

Under the assumption that the data lies on $M = \mathrm{span}\{e_1, \ldots, e_{d-\ell}\}$, and no regularization is used, we can show that the weights of the first layer projected on $M^\perp$ do not change during training. This is an essential part of the proofs, as it allows us to analyze those weights as random Gaussian vectors, and apply concentration bounds on them.

**Theorem A.2.** *Let $M = \mathrm{span}\{e_1, \ldots, e_{d-\ell}\}$. Assume we train a neural network $N(x, \mathbf{w}_{1:m}) := \sum_{i=1}^m u_i \sigma(w_i^\top x)$ as explained in Section 3 (where $\mathbf{w}_{1:m} = (w_1, \ldots, w_m)$). Denote by $\hat{w} := \Pi_{M^\perp}(w)$ for $w \in \mathbb{R}^d$, then after training, for each $i \in [m]$, $\hat{w}_i$ did not change from their initial value.*

*Proof.* Note that for each $i \in [m]$ and $x \in M$ we have:

$$\Pi_{M^\perp}\left(\frac{\partial N(x, \mathbf{w}_{1:m})}{\partial w_i}\right) = \Pi_{M^\perp}\left(u_i \sigma'(w_i^\top x)x\right) = u_i \sigma'(w_i^\top x)\hat{x} = \mathbf{0} \ .$$

Taking the derivative of the loss we have:

$$\Pi_{M^\perp}\left(\frac{\partial L\left(N(x, \mathbf{w}_{1:m}) \cdot y\right)}{\partial w_i}\right) = \Pi_{M^\perp}\left(L'\left(N(x, \mathbf{w}_{1:m}) \cdot y\right) \cdot \frac{\partial N(x, \mathbf{w}_{1:m})}{\partial w_i}\right)$$

$$= L'\left(N(x, \mathbf{w}_{1:m}) \cdot y\right) \cdot \Pi_{M^\perp}\left(\frac{\partial N(x, \mathbf{w}_{1:m})}{\partial w_i}\right) = \mathbf{0} \ .$$

The above calculation did not depend on the specific value of the $w_i$'s. Hence, the value of the $\hat{w}_i$'s for every $i \in [m]$ did not change during training from their initial value. □

# B Proofs from Section 4

Before proving the main theorem, we will first need the next two lemmas about the concentration of Gaussian random variables:

**Lemma B.1.** *Let $w \in \mathbb{R}^n$ such that $w \sim \mathcal{N}(\mathbf{0}, \sigma^2 I_n)$. Then:*

$$\mathbb{P}\left[\|w\|^2 \leq \frac{1}{2}\sigma^2 n\right] \leq e^{-\frac{n}{16}} \ .$$

*Proof.* Note that $\left\|\frac{w}{\sigma}\right\|^2$ has the Chi-squared distribution. A concentration bound by Laurent and Massart [Laurent and Massart, 2000, Lemma 1] implies that for all $t > 0$ we have

$$\Pr\left[n - \left\|\frac{w}{\sigma}\right\|^2 \geq 2\sqrt{nt}\right] \leq e^{-t} \ .$$

Plugging-in $t = \frac{n}{16}$, we get

$$\Pr\left[n - \left\|\frac{w}{\sigma}\right\|^2 \geq \frac{1}{2}n\right] = \Pr\left[\left\|\frac{w}{\sigma}\right\|^2 \leq \frac{1}{2}n\right] \leq e^{-n/16} \ .$$

Thus, we have

$$\Pr\left[\|w\| \leq \sigma\sqrt{\frac{n}{2}}\right] \leq e^{-n/16} \ .$$

□

**Lemma B.2.** *Let $w_1, \ldots, w_m \in \mathbb{R}^n$ such that for all $i \in [m]$, $w_i \sim \mathcal{N}(\mathbf{0}, \sigma^2 I_n)$, then we have:*

$$\mathbb{P}\left[\left\|\sum_{i=1}^m w_i\right\|^2 \leq \frac{1}{2}m\sigma^2 n\right] \leq e^{-\frac{n}{16}} \ .$$

*Proof.* We denote the $j$-th coordinate of the vector $w_i \in \mathbb{R}^n$ by $w_{i,j}$. Note, for any $i \in \{1, \ldots, m\}$ and $j \in \{1, \ldots, n\}$ we have $w_{i,j} \sim \mathcal{N}(0, \sigma^2)$. We denote by $s$ the sum vector $s := \sum_{i=1}^{m} w_i$, and by $s_j$ the $j$-th coordinate of $s$. By this definition, $s_j = \sum_{i=1}^{m} w_{i,j}$ is a sum of $m$ independent Gaussian variables and therefore also a Gaussian variable. Particularly, $s \sim \mathcal{N}(\mathbf{0}, m\sigma^2 I_n)$. We use Lemma B.1 with variance $m\sigma^2$ and get that:

$$\mathbb{P}\left[\left\|\sum_{i=1}^{m} w_i\right\|^2 \leq \frac{1}{2} m\sigma^2 n\right] \leq e^{-\frac{n}{16}} .$$

$\square$

We are now ready to prove the main theorem of this section:

*Proof of Theorem 4.1.* Let $M = \text{span}\{e_1, \ldots, e_{d-\ell}\}$. By Theorem A.1(1), given a training dataset $X \subseteq P$, it is enough to consider a training set $X_R = \{Rx : x \in X\}$, where $R$ is an orthogonal matrix such that $R \cdot P = M$, and training is done over $X_R$. From now on, we assume that the training data, as well as $x_0$ lie on $M$, and the consequences of this proof would also imply for a dataset $X$ and $x_0 \in P$.

The projection of the gradient on $M^\perp$ is equal to:

$$\Pi_{M^\perp}\left(\frac{\partial N(x_0)}{\partial x}\right) = \Pi_{M^\perp}\left(\sum_{i=1}^{m} u_i w_i \mathbb{1}_{\langle w_i, x_0 \rangle \geq 0}\right) = \sum_{i=1}^{m} \Pi_{M^\perp}(u_i w_i) \mathbb{1}_{i \in S} = \sum_{i \in S} \Pi_{M^\perp}(u_i w_i) .$$

Denote by $\hat{w}_i = (w_i)_{d-\ell+1:d}$, the last $\ell$ coordinates of $w_i$. By Theorem A.2 we get that for every $i \in [m]$, $\hat{w}_i$ did not change from their initial value during training.

Recall that we initialized $\hat{w}_i \sim \mathcal{N}(\mathbf{0}, \frac{1}{\sqrt{d}} I_\ell)$. Note that the set $S$ is independent of the value of the $\hat{w}_i$'s. This is because $\hat{w}_i$ does not effect the training, hence will not effect $w_i - \Pi_{M^\perp}(w_i)$. Also, after choosing $x_0$ we have $\langle \hat{w}_i, \hat{x}_0 \rangle = 0$, since $\hat{x}_0 = \mathbf{0}$, which means that the choice of $S$ is independent of the $\hat{w}_i$'s. We can conclude that the random variables $\hat{w}_i$ for $i \in S$ are sampled independently.

Note, since for all $i \in \{1, \ldots, m\}$, $|u_i| = \frac{1}{\sqrt{m}}$ and they are not trained, we get that $u_i \hat{w}_i$ are also Gaussian random variables with the same mean, and variance multiplied by $\frac{1}{m}$. Therefore, from Lemma B.2 we get that w.p. $\geq 1 - e^{-\ell/16}$:

$$\left\|\sum_{i \in S} u_i \hat{w}_i\right\| \geq \sqrt{\frac{1}{2}} \sqrt{\frac{kl}{dm}} .$$

Combining the above, we get:

$$\left\|\Pi_{M^\perp}\left(\frac{\partial N(x_0)}{\partial x}\right)\right\| \geq \sqrt{\frac{1}{2}} \sqrt{\frac{kl}{dm}} .$$

$\square$

## C   Proofs from Section 5

Before proving the main theorem, we prove a few lemmas about concentration of Gaussian random variables:

**Lemma C.1.** *Let $w \in \mathbb{R}^n$ with $w \sim \mathcal{N}(\mathbf{0}, \sigma^2 I_n)$. Then:*

$$\Pr\left[\|w\|^2 \geq 2\sigma^2 n\right] \leq e^{-\frac{n}{16}} .$$

*Proof.* Note that $\left\lVert \frac{w}{\sigma} \right\rVert^2$ has the Chi-squared distribution. A concentration bound by Laurent and Massart [Laurent and Massart, 2000, Lemma 1] implies that for all $t > 0$ we have

$$\Pr\left[ \left\lVert \frac{w}{\sigma} \right\rVert^2 - n \geq 2\sqrt{nt} + 2t \right] \leq e^{-t} .$$

Plugging-in $t = \frac{n}{16}$, we get

$$\Pr\left[ \left\lVert \frac{w}{\sigma} \right\rVert^2 \geq 2n \right] \leq \Pr\left[ \left\lVert \frac{w}{\sigma} \right\rVert^2 - n \geq n/2 + n/8 \right] \leq e^{-n/16} .$$

Thus, we have

$$\Pr\left[ \lVert w \rVert \geq \sigma\sqrt{2n} \right] \leq e^{-n/16} .$$

$\square$

**Lemma C.2.** *Let $u \in \mathbb{R}^n$, and $v \sim \mathcal{N}(\mathbf{0}, \sigma^2 I_n)$. Then, for every $t > 0$ we have*

$$\Pr\left[ |\langle u, v \rangle| \geq \lVert u \rVert\, t \right] \leq 2\exp\left( -\frac{t^2}{2\sigma^2} \right) .$$

*Proof.* We first consider $\langle \frac{u}{\lVert u \rVert}, v \rangle$. As the distribution $\mathcal{N}(\mathbf{0}, \sigma^2 I_n)$ is rotation invariant, one can rotate $u$ and $v$ to get $\tilde{u}$ and $\tilde{v}$ such that $\frac{\tilde{u}}{\lVert u \rVert} = e_1$, the first standard basis vector and $\langle \frac{u}{\lVert u \rVert}, v \rangle = \langle \frac{\tilde{u}}{\lVert u \rVert}, \tilde{v} \rangle$. Note, $v$ and $\tilde{v}$ have the same distribution. We can see that $\langle \frac{\tilde{u}}{\lVert u \rVert}, \tilde{v} \rangle \sim \mathcal{N}(0, \sigma^2)$ since it is the first coordinate of $\tilde{v}$. By a standard tail bound, we get that for $t > 0$:

$$\Pr\left[ |\langle \frac{u}{\lVert u \rVert}, v \rangle| \geq t \right] = \Pr\left[ |\langle \frac{\tilde{u}}{\lVert u \rVert}, \tilde{v} \rangle| \geq t \right] = \Pr\left[ |\tilde{v}_1| \geq t \right] \leq 2\exp\left( -\frac{t^2}{2\sigma^2} \right) .$$

Therefore

$$\Pr\left[ |\langle u, v \rangle| \geq \lVert u \rVert\, t \right] \leq 2\exp\left( -\frac{t^2}{2\sigma^2} \right) .$$

$\square$

**Lemma C.3.** *Let $u \sim \mathcal{N}(\mathbf{0}, \sigma_1^2 I_n)$, and $v \sim \mathcal{N}(\mathbf{0}, \sigma_2^2 I_n)$. Then, for every $t > 0$ we have*

$$\Pr\left[ |\langle u, v \rangle| \geq \sigma_1\sqrt{2n}t \right] \leq e^{-n/16} + 2e^{-t^2/2\sigma_2^2} .$$

*Proof.* Using Lemma C.1 we get that w.p. $\leq e^{-n/16}$ we have $\lVert u \rVert \geq \sigma_1\sqrt{2n}$. Moreover, by Lemma C.2, w.p. $\leq 2\exp\left( -\frac{t^2}{2\sigma_2^2} \right)$ we have $|\langle u, v \rangle| \geq \lVert u \rVert\, t$. By the union bound, we get

$$\Pr\left[ |\langle u, v \rangle| \geq \sigma_1\sqrt{2n}t \right] \leq \Pr\left[ \lVert u \rVert \geq \sigma_1\sqrt{2n} \right] + \Pr\left[ |\langle u, v \rangle| \geq \lVert u \rVert\, t \right] \leq e^{-n/16} + 2\exp\left( -\frac{t^2}{2\sigma_2^2} \right) .$$

$\square$

We are now ready to prove the main theorem of this section:

*Theorem 5.1.* By Theorem A.1(2), we can assume w.l.o.g. that $P = M = \mathrm{span}\{e_1, \ldots, e_{d-\ell}\}$. We also assume w.l.o.g. that $y_0 = 1$, the case $y_0 = -1$ is proved in a similar manner. Denote by $\bar{w} := (w)_{d-\ell+1:d}$, the last $\ell$ coordinates of $w$. By Theorem A.2 we have that $\bar{w}_i$ have not changed after training from their initial value.

We can write $N(x_0 + z)$ as:

$$N(x_0 + z) = \sum_{i=1}^{m} u_i \sigma(\langle w_i, x_0 \rangle + \langle w_i, z \rangle)$$

$$= \sum_{i \in I_-} u_i \sigma(\langle w_i, x_0 \rangle + \langle w_i, z \rangle) + \sum_{i \in I_+} u_i \sigma(\langle w_i, x_0 \rangle + \langle w_i, z \rangle)$$

$$= \sum_{i \in I_-} u_i \sigma(\langle w_i, x_0 \rangle + \langle \bar{w}_i, \bar{z} \rangle) + \sum_{i \in I_+} u_i \sigma(\langle w_i, x_0 \rangle + \langle \bar{w}_i, \bar{z} \rangle) \qquad (2)$$

where the last equality is since $(z)_{1:d-\ell} = \mathbf{0}$, hence $\langle w, z \rangle = \langle \bar{w}, \bar{z} \rangle$ for every $w \in \mathbb{R}^d$. We will bound each term of the above separately.

For the first term in Eq. (2), where $i \in I_-$ we can write:

$$\langle \bar{w}_i, \bar{z} \rangle = \alpha \left\| \bar{w}_i \right\|^2 + \alpha \langle \bar{w}_i, \sum_{j \neq i} \text{sign}(u_j) \bar{w}_j \rangle .$$

By our assumptions, $\bar{w}_i \sim \mathcal{N}\left(\mathbf{0}, \frac{1}{d} I_\ell\right)$ and $\sum_{j \neq i} \text{sign}(u_j) \bar{w}_j \sim \mathcal{N}\left(\mathbf{0}, \frac{m-1}{d} I_\ell\right)$, since it is a sum of $m - 1$ i.i.d. Gaussian random variables, which are also symmetric hence multiplying them by $-1$ does not change their distribution. From Lemma B.1 we get w.p. $\geq 1 - e^{-\ell/16}$ that

$$\alpha \cdot \left\| \bar{w}_i \right\|^2 \geq \alpha \cdot \frac{\ell}{2d} .$$

From Lemma C.3, and using $t = \sqrt{\frac{(m-1)\log(dm^2)}{d}}$ we get w.p. $\geq 1 - e^{-\ell/16} + 2e^{-t^2 d/2(m-1)} = 1 - e^{-\ell/16} + 2m^{-1} d^{-1/2}$ that

$$\langle \bar{w}_i, \sum_{j \neq i} \text{sign}(u_j) \bar{w}_j \rangle \leq \frac{1}{\sqrt{d}} t \sqrt{2\ell}$$

$$= \frac{1}{d} \cdot \sqrt{2\ell(m-1)\log(m^2 d)} . \qquad (3)$$

Applying union bound over the above two events, and for every $i \in I_-$, we get w.p. $\geq 1 - 2\left(me^{-\ell/16} + d^{-1/2}\right)$ that:

$$\langle \bar{w}_i, \bar{z} \rangle \geq \frac{\alpha \ell}{2d} - \frac{\alpha}{d} \sqrt{2\ell(m-1)\log(m^2 d)} .$$

For the second term in Eq. (2), where $i \in I_+$ we can write in a similar way:

$$\langle \bar{w}_i, \bar{z} \rangle = -\alpha \left\| \bar{w}_i \right\|^2 + \alpha \langle \bar{w}_i, \sum_{j \neq i} \text{sign}(u_j) \bar{w}_j \rangle .$$

Using the same argument as above, we get w.p $\geq 1 - 2\left(me^{-\ell/16} + d^{-1/2}\right)$ that:

$$\langle \bar{w}_i, \bar{z} \rangle \leq -\frac{\alpha \ell}{2d} + \frac{\alpha}{d} \sqrt{2\ell(m-1)\log(m^2 d)} .$$

By assuming that $\ell \geq 8(m-1)\log(m^2 d)$ we get that $\langle \bar{w}_i, z \rangle \leq 0$. Denote $C := \frac{\alpha \ell}{2d} - \frac{\alpha}{d}\sqrt{2\ell(m-1)\log(m^2 d)}$, then going back to Eq. (2), using the above bounds and applying union bound, we get w.p. $\geq 1 - 4\left(me^{-\ell/16} + d^{-1/2}\right)$ that:

$$N(x_0 + z) \leq \sum_{i \in I_-} u_i \sigma(\langle w_i, x_0 \rangle + C) + \sum_{i \in I_+} u_i \sigma(\langle w_i, x_0 \rangle)$$

$$= \sum_{i \in I_-} u_i \sigma(\langle w_i, x_0 \rangle + C) + \sum_{i \in I_+} u_i \sigma(\langle w_i, x_0 \rangle) + \sum_{i \in I_-} u_i \sigma(\langle w_i, x_0 \rangle) - \sum_{i \in I_-} u_i \sigma(\langle w_i, x_0 \rangle)$$

$$= \sum_{i \in I_-} u_i \sigma(\langle w_i, x_0 \rangle + C) - \sum_{i \in I_-} u_i \sigma(\langle w_i, x_0 \rangle) + N(x_0)$$

$$= \sum_{i \in I_-} u_i \left(\sigma(\langle w_i, x_0 \rangle + C) - \sigma(\langle w_i, x_0 \rangle)\right) + N(x_0) .$$

Define $F_- := \{i \in I_- : \langle w_i, x_0 \rangle \geq 0\}$, and $k_- = |F_-|$. We have that:

$$\sum_{i \in I_-} u_i \left( \sigma(\langle w_i, x_0 \rangle + C) - \sigma(\langle w_i, x_0 \rangle) \right) \leq \sum_{i \in F_-} u_i \left( \sigma(\langle w_i, x_0 \rangle + C) - \sigma(\langle w_i, x_0 \rangle) \right)$$

$$= \sum_{i \in F_-} u_i C = -\frac{k_- C}{\sqrt{m}} \ ,$$

where the first inequality is since we only sum over negative terms, and the second inequality is since both $\langle w_i, x_0 \rangle \geq 0$ (because $i \in F_-$) and $C \geq 0$ (because $\ell \geq 32(m-1)\log(m^2 d)$). Combining all of the above, we get that:

$$N(x_0 + z) \leq -\frac{k_- C}{\sqrt{m}} + N(x_0) \ . \tag{4}$$

By our assumption that $\ell \geq 32(m-1)\log(m^2 d)$ we have that

$$C = \alpha \left( \frac{1}{2} \frac{\ell}{d} - \sqrt{2}\sqrt{m-1} \frac{\sqrt{\ell}}{d} \sqrt{\log(dm^2)} \right)$$

$$= \frac{\alpha \sqrt{\ell}}{d} \left( \frac{\sqrt{\ell}}{2} - \sqrt{2(m-1)\log(m^2 d)} \right)$$

$$\geq \frac{\alpha \ell}{4d} \ .$$

Plugging in $C$ and $\alpha = \frac{8\sqrt{m}dN(x_0)}{k_- \ell}$ to Eq. (4) we get that:

$$N(x_0 + z) \leq -\frac{k_- C}{\sqrt{m}} + N(x_0)$$

$$\leq -\frac{k_-}{\sqrt{m}} \cdot \frac{\ell}{4d} \cdot \frac{8\sqrt{m}dN(x_0)}{k_- \ell} + N(x_0) = -N(x_0) < 0 \ ,$$

and in particular $\text{sign}(N(x_0)) \neq \text{sign}(N(x_0 + z))$.

We are left with calculating the norm of $z$:

$$\|z\| = \alpha \cdot \left\| \sum_{i \in I_-} \Pi_{M^\perp}(w_i) - \sum_{i \in I_+} \Pi_{M^\perp}(w_i) \right\|$$

$$= \alpha \cdot \left\| \sum_{i=1}^{m} -\text{sign}(u_i)\Pi_{M^\perp}(w_i) \right\|$$

$$= \alpha \cdot \left\| \sum_{i=1}^{m} -\text{sign}(u_i)\bar{w}_i \right\| \ .$$

Since for each $i \in [m]$, $\bar{w}_i \sim \mathcal{N}\left(\mathbf{0}, \frac{1}{d}I_\ell\right)$, then $-\text{sign}(u_i)\bar{w}_i$ also have the same distribution, because this is a symmetric distribution. Hence, $\sum_{i=1}^{m} -\text{sign}(u_i)\bar{w}_i \sim \mathcal{N}\left(\mathbf{0}, \frac{m}{d}I_\ell\right)$ as a sum of Gaussian random variables. Using Lemma C.1 we get w.p $\geq 1 - e^{-\ell/16}$ that $\|\sum_{i=1}^{m} -\text{sign}(u_i)\bar{w}_i\|^2 \leq \frac{2m\ell}{d}$. Plugging in $\alpha$ we get that:

$$\|z\| \leq \sqrt{\frac{2m\ell}{d}} \cdot \frac{8\sqrt{m}dN(x_0)}{k_- \ell} = 8\sqrt{2}N(x_0) \cdot \frac{m}{k_-} \cdot \sqrt{\frac{d}{\ell}} \ .$$

$\square$

# D  Proofs for Section 6

For proving the main theorem, we will use the following lemma that upper bounds the norm of a sum of Gaussian random variables:

**Lemma D.1.** *Let $w_1, .., w_m \in \mathbb{R}^n$ such that for all $i \in [m]$, $w_i \sim \mathcal{N}(\mathbf{0}, \sigma^2 I_n)$, then we have:*

$$\mathbb{P}\left[\left\|\sum_{i=1}^m w_i\right\|^2 \geq 2m\sigma^2 n\right] \leq e^{-\frac{n}{16}}$$

*Proof.* We denote the $j$-th coordinate of the vector $w_i \in \mathbb{R}^n$ by $w_{i,j}$. Note, for any $i \in [m]$ and $j \in [n]$ we have $w_{i,j} \sim \mathcal{N}(0, \sigma^2)$. We denote by $s$ the sum vector $s := \sum_{i=1}^m w_i$, and by $s_j$ the $j$-th coordinate of $s$. By this definition, $s_j = \sum_{i=1}^m w_{i,j}$ is a sum of $m$ independent Gaussian variables and therefore also a Gaussian variable. Therefore, $s \sim \mathcal{N}(\mathbf{0}, m\sigma^2 I_n)$. We use Lemma C.1 with variance $m\sigma^2$ and get that:

$$\mathbb{P}\left[\left\|\sum_{i=1}^m w_i\right\|^2 \geq 2m\sigma^2 n\right] \leq e^{-\frac{n}{16}}.$$

$\square$

We now prove the main theorem of this section:

*Proof of Theorem 6.1.* Similar to the lower bound of the norm, let $M = \mathrm{span}\{e_1, \ldots, e_{d-\ell}\}$. By Theorem A.1(1), given a training dataset $X \subseteq P$, it is enough to consider a training set $X_R = \{Rx : x \in X\}$, where $R$ is an orthogonal matrix such that $R \cdot P = M$, and training is done over $X_R$. From now on, we assume that the training data, as well as $x_0$ lie on $M$, and the consequences of this proof would also imply for a dataset $X$ and $x_0 \in P$.

The projection of the gradient on $M^\perp$ is equal to:

$$\Pi_{M^\perp}\left(\frac{\partial N(x_0)}{\partial x}\right) = \Pi_{M^\perp}\left(\sum_{i=1}^m u_i w_i \mathbb{1}_{\langle w_i, x_0\rangle \geq 0}\right) = \sum_{i=1}^m \Pi_{M^\perp}(u_i w_i)\,\mathbb{1}_{i \in S} = \sum_{i \in S}\Pi_{M^\perp}(u_i w_i).$$

Denote by $\hat{w}_i = (w_i)_{d-\ell+1:d}$, the last $\ell$ coordinates of $w_i$. By Theorem A.2 we get that for every $i \in [m]$, $\hat{w}_i$ did not change from their initial value during training.

Recall that we initialized $\hat{w}_i \sim \mathcal{N}(\mathbf{0}, \beta^2 I_\ell)$. Note that the set $S$ is independent of the value of the $\hat{w}_i$'s. This is because $\hat{w}_i$ does not effect the training, hence will not effect $w_i - \Pi_{M^\perp}(w_i)$. Also, after choosing $x_0$ we have $\langle \hat{w}_i, \hat{x}_0\rangle = 0$, since $\hat{x}_0 = \mathbf{0}$, which means that the choice of $S$ is independent of the $\hat{w}_i$'s. We can conclude that the random variables $\hat{w}_i$ for $i \in S$ are sampled independently.

Therefore, from Lemma B.2 we get that w.p. $\geq 1 - e^{-\ell/16}$:

$$\left\|\sum_{i \in S}\hat{w}_i\right\| \leq \beta\sqrt{2k\ell}.$$

Note, since for all $i \in [m]$, $|u_i| = \frac{1}{\sqrt{m}}$ and they are not trained, we get w.p. $\geq 1 - e^{-\ell/16}$ that:

$$\left\|\Pi_{M^\perp}\left(\frac{\partial N(x_0)}{\partial x}\right)\right\| \leq \beta\sqrt{\frac{2k\ell}{m}}.$$

$\square$

### D.1 Explicit $L_2$ regularization

*Proof of Theorem 6.2.* As before, for this proof we rotate the data subspace $P$ to lie on $M = \text{span}\{e_1, \ldots, e_{d-\ell}\}$ and rotate the model's weights accordingly. For a dataset $(x_1, y_1), .., (x_r, y_r)$, we train over the following objective:

$$\sum_{j=1}^{r} L(y_j \cdot N(x_j, \mathbf{w}_{1:m}))) + \frac{1}{2}\lambda \|\mathbf{w}_{1:m}\|^2$$

In Theorem A.2, we showed for all $(x_j, y_j)$ that if we train the model using the loss $L$ we get:

$$\Pi_{M^\perp} \left( \frac{\partial L\left(N(x_j, \mathbf{w}_{1:m}) \cdot y_j\right)}{\partial w_i} \right) = 0$$

Now, we analyze the training process using the new loss which includes the regularization term. We denote by $w_i^{(t)}$ the weight vector $w_i$ after $t$ training steps, and by $\hat{w}_i^{(t)} := \Pi_{M^\perp}\left(w_i^{(t)}\right)$ its projection on the subspace orthogonal to $M$. We look at the projected gradient of $w_i^{(t)}$ w.r.t. the loss:

$$\Pi_{M^\perp} \left( \frac{\partial \sum_{j=1}^r L\left(N(x_j, \mathbf{w}_{1:m}^{(t)}) \cdot y_j\right)}{\partial w_i} + \frac{\partial \frac{1}{2}\lambda \left\|w_i^{(t)}\right\|^2}{\partial w_i} \right) =$$

$$= \sum_{j=1}^{r} \Pi_{M^\perp} \left( \frac{\partial L\left(N(x_j, \mathbf{w}_{1:m}^{(t)}) \cdot y_j\right)}{\partial w_i} \right) + \Pi_{M^\perp} \left( \frac{\partial \frac{1}{2}\lambda \left\|w_i^{(t)}\right\|^2}{\partial w_i} \right)$$

$$= \Pi_{M^\perp} \left( \frac{\partial \frac{1}{2}\lambda \left\|w_i^{(t)}\right\|^2}{\partial w_i} \right)$$

$$= \Pi_{M^\perp} \left( \lambda w_i^{(t)} \right)$$

$$= \lambda \hat{w}_i^{(t)}.$$

For a training step of size $\eta$, using gradient descent we get that:

$$\hat{w}_i^{(t+1)} = \hat{w}_i^{(t)} - \eta\lambda\hat{w}_i^{(t)}.$$

Thus, after a total of $T$ iteration of training we get that:

$$\hat{w}_i^{(T)} = (1 - \eta\lambda)^T \hat{w}_i^{(0)}.$$

Therefore, the projection of gradients after training onto $P^\perp$ will be the same as if they were initialized to $\sim \mathcal{N}\left(0, \frac{(1-\eta\lambda)^{2T}}{d} I_d\right)$ and trained using logistic loss without regularization. The rest of the proof is the same as Theorem 6.1 for $\beta = \frac{(1-\eta\lambda)^T}{\sqrt{d}}$. $\qquad\square$

## E   Further Experiments and Experimental Details

### E.1   Further Experiments

In Figure 5 we present the boundary of a two-layer ReLU network trained over a 25-point dataset on a two-dimensional linear subspace, similar to Figure 3. We train the networks until reaching a constant positive margin. The difference between the figures is that in Figure 5 we initialize the weights using the default PyTorch initialization, while in Figure 3 we initialized using a smaller scale

for the robustness effect to be smaller, and visualized more easily. The experiment in Figure 5 is demonstrating an extreme robustness effect, occurring when using the standard settings.

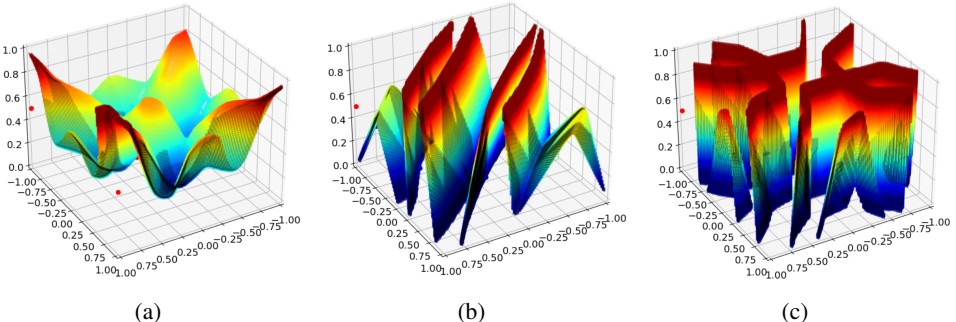

|     |     |     |
|:---:|:---:|:---:|
| (a) | (b) | (c) |

Figure 5: **Experiments on two-dimensional dataset.** We plot the dataset points and the decision boundary in 3 settings: (a) Vanilla trained network, (b) The network's weights are initialized from a smaller variance distribution, and (c) Training with regularization. Colors are used to emphasise the values in the $z$ axis.

In Figure 6 we go beyond the theory discussed in this paper, and present similar phenomena in all three settings for a five-layer ReLU network. In Figure 6a we can see the boundary of the regularly trained network within a small distance in $P^\perp$ from the data points. In Figure 6b we use small initialization for all five layers, and present a boundary almost orthogonal to the data manifold. In Figure 6c, the boundary of a regularized trained network is in a similar form. This experiment suggests that our theoretical results might be extended also to deeper networks, where all layers are trained.

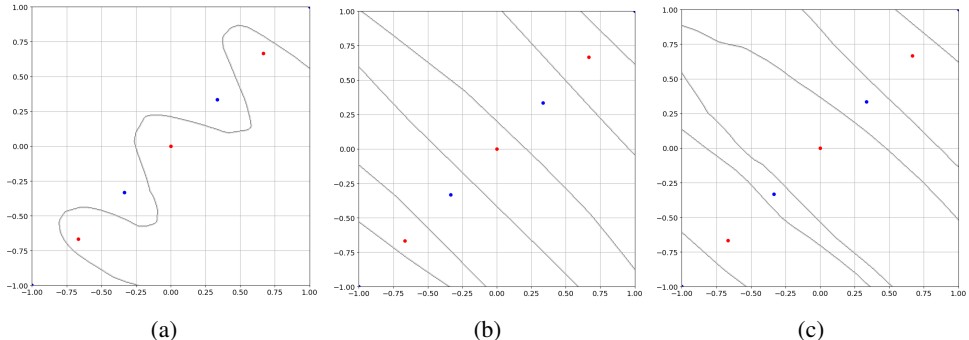

|     |     |     |
|:---:|:---:|:---:|
| (a) | (b) | (c) |

Figure 6: **Experiments on one-dimensional dataset with deep network.** We plot the dataset points and the decision boundary in 3 settings: (a) Vanilla trained network, (b) The network's weights are initialized from a smaller variance distribution, and (c) Training with regularization.

### E.2  One-dimensional dataset experiment - 2 layer network (Figure 2)

**Dataset**  For all the three experiments we used a 7-point data set, spread equally on the two dimensional line $y = x$ from $(-1, -1)$ to $(1, 1)$.

**Network**  For all the three experiments we used two-layer ReLU network of width 100 with biases in both layers. The weights of both layers were initialized using (1+3) default PyTorch initialization for linear layers, (2) default initialization divided by 3.

**Training**  We used train step of size $0.02$ for (1+3) and $0.04$ for (2). We trained both layers until the margin reached $0.3$. The losses we used were (1+2) Logistic loss, (3) Logistic loss with $0.005$ $L_2$ regularization.

### E.3 Two-dimensional dataset experiment - smaller effect (Figure 3)

**Dataset**  For all the three experiments we used a 25-point data set, spread equally on a grid which lies on the $z = 0.5$ axis.

**Network**  For all the three experiments we used two-layer ReLU network of width $4000$ with biases in both layers. The weights in the first layer were initialized in (1+3) from $\mathcal{N}(\mathbf{0}, 1/3I_3)$, and in (2) from $\mathcal{N}(\mathbf{0}, 1/36I_3)$. The weight of the output layer were initialized to the uniform distribution over the set $\{-1, 1\}$.

**Training**  For all the experiments we trained both layers until the margin reached 0.3 and we used train step of size 0.002. The losses we used were (1+2) Logistic loss, (3) Logistic loss with 0.8 $L_2$ regularization on the weights of the first layer.

### E.4 Two-dimensional dataset experiment (Figure 5)

**Dataset**  For all the three experiments we used a 25-point data set, spread equally on a grid which lies on the $x - y$ axis.

**Network**  For all the three experiments we used two-layer RelU network of width $400$ with biases in both layers. The weights in any layer were initialized using (1+3) default PyTorch initialization for linear layers, (2) default initialization divided by 3.

**Training**  For (1) experiments we used train step of size 0.005, and for (2+3) we used step of size 0.05. We trained both layers until the margin reached 0.1. The losses we used were (1+2) Logistic loss, (3) Logistic loss with 0.005 $L_2$ regularization.

### E.5 One-dimensional dataset experiment - 5 layer network (Figure 6)

**Dataset**  For all the three experiments we used a 7-point data set, spread equally on the two dimensional line $y = x$ from $(-1, -1)$ to $(1, 1)$.

**Network**  For all the three experiments we used 5-layer RelU network of width $100$ with biases in all layers. The weights in any layer were initialized using (1+3) default PyTorch initialization for linear layers, (2) default initialization divided by 3.

**Training**  For (1+3) experiments we used train step of size 0.02, and for (2) we used step of size 0.06. we trained all layers until the margin reached 0.3. The losses we used were (1+2) Logistic loss, (3) Logistic loss with 0.01 $L_2$ regularization.

