# OpenReview forum: "Adversarial Examples Exist in Two-Layer ReLU Networks for Low Dimensional Linear Subspaces"
_NeurIPS.cc/2023/Conference — NeurIPS 2023 poster_

### Official Review · Reviewer_2uZr · 2023-06-14

**Soundness:** 4 excellent
**Presentation:** 4 excellent
**Contribution:** 3 good
**Rating:** 6
**Confidence:** 4

**Summary:**

The paper investigates adversarial robustness of trained ReLU neural networks when data lies within a linear subspace.  For the theoretical part, which is the bulk of the paper, the networks have two layers and only the first layer is trained.  Then the key observation is that the assumption on the data causes the training to change only the projections of the first-layer weight vectors to the linear subspace, i.e. their components in the orthogonal subspace remain as at initialisation.  Based on this, the authors prove several results, including a lower bound on the projection to the orthogonal subspace of the gradient of the network output at any point in the linear subspace, existence of a direction in the orthogonal subspace for universal adversarial perturbations, and that either reducing the initialisation scale or using L2 regularisation can reduce the norm of the projections to the orthogonal subspace of the gradients (in the latter case, the projections of the first-layer weight vectors to the orthogonal subspace are changed during training only by the L2 regulariser).  The results contains some assumptions, most notably for the universal adversarial perturbation, the width of the network has to be roughly at most a constant fraction of the input dimension.  The theory is supplemented by experiments on small examples in dimensions 2 and 3; in them all layers are trained and a network with five layers is also considered, which suggests that it may be possible to extend the theoretical results beyond the assumptions in the paper.


**Strengths:**

The paper is a nice mostly theoretical follow-up to the mostly empirical "The Dimpled Manifold Model of Adversarial Examples in Machine Learning" by Shamir, Melamed, and BenShmuel (CoRR abs/2106.10151), which in particular brings in the considerations of the effects of initialisation scale and L2 regularisation.  The theorems are proved in detail in the appendix, with helpful sketches provided in the main.  The experiments and the resulting figures aid the intuition, and suggest directions for future theoretical work.


**Weaknesses:**

The proofs of the theoretical results are perhaps not surprising or difficult.

I believe the code for the experiments is relatively simple and essentially given by the text in Appendix E, however it was not submitted with the supplementary materials.


**Questions:**

Are you able to say how the theoretical results would change if the second-layer weights were trained as well?


**Limitations:**

The assumption of small network width in relation to the input dimension is restrictive, although a similar assumption featured in the related paper by Daniely and Shacham (NeurIPS 2020).

---

> ### Author Rebuttal · Authors · 2023-08-09
>
> We thank the reviewer for the thorough review and constructive comments.
>
> Code for the experiments: We indeed report all the experimental details in Appendix E. We will publish the full code with the camera-ready version.
>
> "Are you able to say how the theoretical results would change if the second-layer weights were trained as well?": This is a good question and an interesting future research direction. We believe it is possible to show a similar result when the second layer weights are also trained. However, such a result would require stronger assumptions on the input data, beyond residing on a low dimensional manifold. Therefore we tried to avoid it in our paper and keep the conclusions as general as possible. Empirically, we trained all layers and showed similar results in the experiments attached here and in the paper.

---

> > ### Comment · Reviewer_2uZr · 2023-08-10
> >
> > Thank you for these responses.

---

### Official Review · Reviewer_X8ox · 2023-06-15

**Soundness:** 3 good
**Presentation:** 3 good
**Contribution:** 3 good
**Rating:** 6
**Confidence:** 4

**Summary:**

This paper studies the vulnerability of two-layer ReLU networks under adversarial attack when the data is in a low-dimensional manifold. The paper also observes that adding L2 regularization in clean training also improves the adversarial robustness.

**Strengths:**

The intuition of this paper is interesting, with a high quality and a good clarity. Although existing literature considers that the vulnerability of clean training is caused by the off-manifold attack, this is the first one to study the theoretical perspective in detail.

**Weaknesses:**

(1) My major concern is that this paper mainly studies the side off the data manifold, but not the side on the data manifold. Some claims need more justifications. For example, in the abstract, the authors mention that "the standard gradient methods lead to non-robust neural networks, .. and are susceptible to small adversarial L2 perturbations in these directions". To claim that the perturbation is "small", the authors need to provide a result on what is the smallest on-manifold attack strength which results in an incorrect prediction. It is essential to make comparison in order to claim a quantity is "small", i.e., while Theorem 4.1 presents the L2 norm of the off-manifold gradient, what is the on-manifold gradient?

(2) For the numerical experiments, please also share the code or report the observations when the training iteration T is large enough so that the loss converges to zero. In the paper

Ba, J., Erdogdu, M., Suzuki, T., Wu, D., & Zhang, T. (2020, April). Generalization of two-layer neural networks: An asymptotic viewpoint. In International conference on learning representations.

it is observed that training a neural network in regression in their setting is the same as a penalized linear regression, i.e., the training already introduces some sort of regularization. It would be great if the authors of this submission can provide the code or some more experimental results to support the claims in this submission.

Due to concern (1) and (2), my current score is 5. I like the intuition of this paper so I will consider raising my score if these major concerns can be addressed properly. For (1), a full theoretical proof may takes too much time. A brief justification would be helpful enough.

Some other comments:

(3) While Theorem 6.2 provides justifications on how L2 regularization improves adversarial robustness, it would be helpful if the authors can also provide the convergence and the generalization performance of the regularized neural network, which may potentially implies a trade-off between clean performance and adversarial robustness.


**Questions:**

Please address my concerns mentioned in the weakness section.

---

> ### Author Rebuttal · Authors · 2023-08-09
>
> We thank the reviewer for the thorough review and helpful comments.
>
> Weaknesses:
>
> 1) We emphasize that an analysis of the perturbations on the data manifold would require having some additional assumptions on the data, e.g. about its distribution. We believe that one of the major benefits of our work is that the results are general in the sense that the only assumption we have on the data is that it resides on a low-dimensional subspace. Hence, we can only analyze the gradient and the adversarial perturbations off the manifold. We think it is of great interest to study on-manifold perturbations and compare them to those off-manifold. However, this is beyond the scope of our paper.
>
> 2) We will include all the code for the experiments in the camera-ready version.
>
>     The visualizations in the paper demonstrate the evolution of the decision boundary, rather than the loss of the predictor on the train/test data.
>     Therefore we measured convergence using minimal margin criteria, as noted in Appendix E.
>     We added several new experiments which demonstrate the theoretical phenomenon described in the paper, on both synthetic and real datasets. In these experiments, we trained the network until reaching a cross-entropy loss of $10^{-3}$.
>     Full details about the experiments can be found in our global comment above, and the results are in the attached PDF.
>
> 3) We agree it would be helpful to provide the convergence and generalization performance of the trained predictors and potentially observe a trade-off between robustness and performance. However, to analyze these, we would need further assumptions on the data beyond residing on a low-dimensional manifold. This is because, different data distributions (on the manifold) would mean different generalization or convergence guarantees. We think that a very interesting follow-up work would be to study specific on-manifold distributions, and use the tools we developed in this paper to compare on- and off-manifold robustness with generalization and convergence.

---

> > ### Comment · Reviewer_X8ox · 2023-08-10
> > **Comment**
> >
> > I appreciate the authors providing the more real data experiments in the pdf file. For my review comments, I'm still not convinced:
> >
> > 1. I agree on-manifold perturbation requires extra assumption, but **an example of such a result is necessary to support the claim that "off-manifold perturbation is small"**. (If I say US is closed to Australia, I mean comparing to the distance from the Earth to the sun, but not comparing the distance from New York to California.)
> >
> > 2. Could you summarize the observations of the additional simulations in the comment?

---

> > > ### Author Response · Authors · 2023-08-11
> > > **Re: Comment**
> > >
> > > Thank you for the reply.
> > > Below we provide three arguments on why we consider the off-manifold perturbations small:
> > >
> > > 1) As we elaborated in lines 186-191 and 249-253, if we consider data where each coordinate is of size $\Theta(1)$, then the norm of each data point is $\Theta(\sqrt{d})$ (where $d$ is the input's dimension). In our results, we show that the off-manifold perturbations are much smaller, namely, typically it is in the order of $\text{polylog}(d)$. The size $\Theta(\sqrt{d})$ is a good reference point for comparison, since it is the trivial upper bound for the size of on-manifold perturbations (as we can always flip the output's sign by moving to a point with an opposite label). We note the previous works on adversarial perturbations in random neural networks also used the inputs' norms as the reference point for measuring whether a perturbation is small [1,2,3,4]. Thus, comparing the perturbation's size to the inputs' norms is already common in the relevant literature.
> > >
> > > 2) Empirically, this can be observed in the new experiments we added (please see the attached PDF in the comment to all reviewers, Figure 2(a)). Specifically, for the synthetic experiment we considered where the data lies on a low-dimensional sphere, the size of the off-manifold perturbation for a network trained with standard initialization ($1.0$ in the x-axis) is roughly half of the on-manifold perturbation. This indicates that without small initialization (or weight decay), in certain situations the off-manifold perturbation is smaller than the on-manifold.
> > >
> > > 3) It is easy to construct a contrived example of a data distribution where the on-manifold perturbation is arbitrarily large compared to the off-manifold perturbation. Consider a one-dimensional data manifold, and a dataset consisting of a single data point sampled from this manifold with norm $A$. Suppose also that the network consists of a single neuron (for ease of analysis). In that case, training the network on this dataset until reaching some fixed margin, will result in the neuron being a sum of two components: (1) A component in the direction of the data point; and (2) an orthogonal component to the point, with norm depending on the magnitude of the initialization. An on-manifold perturbation will move only in the direction of the point, hence flipping the label would require a perturbation of magnitude $A$ which moves to the opposite halfspace. On the other hand, an off-manifold perturbation is independent of $A$. We will be happy to elaborate more on this theoretical example if required.
> > >
> > >
> > > Regarding the observations of the additional experiments. We conduct two sets of experiments:
> > >
> > > 1) We perform PCA on MNIST and CIFAR10, and show that most of the variance lies only on few dimensions. Thus, further motivating our assumption that the data lies on a low-dimensional manifold.
> > >
> > > 2) On both a synthetic dataset and MNIST, we compare the off-manifold to on-manifold and overall (i.e both on- and off-manifold) perturbations for networks trained with different weight initializations. The experiments show that smaller initialization increases the off-manifold and overall robustness (by increasing the required perturbation size required for changing a label), while having almost no effect on the on-manifold robustness.
> > >
> > >
> > > [1] Amit Daniely and Hadas Shacham. Most relu networks suffer from ℓ2 adversarial perturbations, 2020
> > >
> > > [2] Sébastien Bubeck, Yeshwanth Cherapanamjeri, Gauthier Gidel, and Rémi Tachet des Combes. A single gradient step finds adversarial examples on random two-layers neural networks, 2021
> > >
> > > [3] Peter Bartlett, Sébastien Bubeck, and Yeshwanth Cherapanamjeri. Adversarial examples in multi-layer random relu networks, 2021
> > >
> > > [4] Andrea Montanari and Yuchen Wu. Adversarial examples in random neural networks with general activations, 2022

---

> > > > ### Comment · Reviewer_X8ox · 2023-08-12
> > > >
> > > > Thanks for the response. The picture is more clear to me now and I have raised the score from 5 to 6.
> > > >
> > > > One additional question:
> > > >
> > > > In Theorem 4.2 of
> > > >
> > > > Frei, Spencer, et al. "The Double-Edged Sword of Implicit Bias: Generalization vs. Robustness in ReLU Networks." arXiv preprint arXiv:2303.01456 (2023).
> > > >
> > > > They use clean training to train both layers of the neural network and also show that the neural network is vulnerable to adversarial attack. If we simply padding zeros to their data to add more dimensions, intuitively the on-manifold attack in their model will be $O(\sqrt{d-l})$ using your notation (I just ignore their denominator). May I know under what conditions in your theorem, the off-manifold attack is stronger than their on-manifold attack? This comparison is not rigorous as it the two settings do not align with each other, but is still helpful to get the picture of the results.

---

> > > > > ### Author Response · Authors · 2023-08-13
> > > > >
> > > > > Thanks!
> > > > >
> > > > > A comparison to Frei et al. is indeed interesting (although, as you mentioned, there are differences between the settings so such a comparison is not rigorous).
> > > > > Suppose that the first $d-l$ coordinates follows the distribution from Frei et al. and the last $l$ coordinates are zero. In their setting, the distance between points of opposite labels is roughly $\sqrt{d-l}$, but they show that there exist adversarial perturbations of size smaller than that. Specifically, in Example 2 they obtain a setting where the perturbation’s size is $\tilde{O}\left((d-l)^{1/4}\right)$. By Theorem 5.1 in our paper, for an input $x_0$ we get an off-manifold adversarial perturbation of size $O \left( N(x_0) \sqrt{\frac{d}{l}} \right)$ (assuming that the number of active neurons is a constant fraction of the number of neurons). By Remark 3.1, we typically have $N(x_0) \leq \text{polylog}(d)$, so overall we get an off-manifold perturbation of size $\tilde{O}\left( \sqrt{\frac{d}{l}}  \right)$. For example, if $l = \Theta(\sqrt{d})$ we get an off-manifold perturbation of size similar to their on-manifold perturbation, but if $l = \Theta(d)$ we get a perturbation of size $\tilde{O}(1)$, which is much smaller than theirs.

---

> > > > > > ### Comment · Reviewer_X8ox · 2023-08-13
> > > > > >
> > > > > > Thanks for the response.

---

### Official Review · Reviewer_5bBJ · 2023-07-02

**Soundness:** 3 good
**Presentation:** 3 good
**Contribution:** 2 fair
**Rating:** 6
**Confidence:** 3

**Summary:**

This paper focus on the data lies on a low dimensional data manifold (linear subspace P). There’re no additional assumptions on the number of data and their structure (orthogonality). The paper considers the perturbation on P^\orth space. The paper claims that standard gradient descent leads to non-robust neural network; while decreasing the initialization scale, adding L2 regularizer can make the trained network more robust.


**Strengths:**

1. Studying the adversarial perturbation on the low-dimensional subspace is an interesting problem.
2. The paper is well-written and easy to follow. The mathematical derivation seems sound.


**Weaknesses:**

1. This paper specifically assumes the dataset lies on a linear subspace, and the perturbation lies in the orthogonal direction of the subspace. The analysis of thm4.1 still relies on part of the weight being unchanged after training, and the gradient lower bound depends on the unchanged weights. To my understanding, such analysis still relies on random initialization property and therefore not much different compared to previous work, which might previous work analysis also holds under this paper’s low dimensionality assumption. To that end, the perturbation lies in the orthogonal direction of the subspace this constraint is trying to bypass the gradient update algorithm. In other words, does it mean that theorem 4.1 still holds even if you consider adversarial training?

2. Although the author motivates the idea by saying real-world dataset mostly lie in low-dimensional subspace, the experiments are based on extremely simple synthetic data. I’m wondering whether similar results hold for a simple dataset like MNIST.

**Questions:**

1. In Theorem 5.1 or Corollary 5.1, the perturbation size depends on the output size. However, in classification problems, you can always rescale the predictor ($w,u$) while maintaining the sign of prediction to be unchanged. To that end, $N(x_0)$ can be arbitrarily small, implying the size of perturbation can be arbitrarily small, regardless of the size of $k_{y_0}$ and $\ell$. Then the result is a bit counterintuitive.

2. I'm wondering whether theorem 4.1 lower bound can also depend on the network initialization so that it matches theorem 6.1.

---

> ### Author Rebuttal · Authors · 2023-08-09
>
> We thank the reviewer for the thorough review and helpful comments.
>
> Regarding the weaknesses:
>
> 1) We acknowledge in the paper that some of our proof techniques were also used in previous works about robustness. The main difference is that in this work we consider trained networks, and our results are general, with only very minor assumptions on the data, while previous works considered random and untrained networks. While the weights corresponding with the off-data dimensions are indeed unchanged, the other weights are updated and affect those gradients via activations. Therefore, even if we use stronger data constraints, results based on random networks do not apply here.  Analyzing adversarial training will require additional assumptions on the input data, this goes beyond the scope of the paper and is an interesting future research direction.  If we didn't understand the question correctly, we would be happy if the reviewer could clarify it.
>
> 2) To show our results indeed extend to the MNIST dataset, we use simple PCA decomposition that demonstrates the implicit low linear data subspace. One can see a small fraction of the dimensions capture most of the variance. We also added an additional experiment on the MNIST dataset demonstrating the phenomena presented in the paper. Details about the experiments can be found in our global comment above, and the results are in the attached PDF.
>
> Regarding the questions:
>
> 1) Theorem 5.1 and Corollary 5.1 indeed depend on the size of the output, but also on the scale of the initialization. Please note that scaling the output (by scaling the weights) in these results is equivalent to scaling the size of the initialization, which, as we show, affects the size of the adversarial perturbation.
>
> 2) This is a good question. Yes, Theorem 4.1 could have been written where the scale of the initialization is a parameter of the problem and then the upper and lower bounds in Theorem 4.1 and Theorem 6.1 would have been tight (up to a constant factor). We wrote the theorem this way to emphasize the phenomenon in standard initializations, and will consider changing it for the camera-ready version.

---

> > ### Comment · Reviewer_5bBJ · 2023-08-11
> >
> > I thank the author for addressing my concerns and presenting a new experiment. I have some followup questions.
> >
> > Regarding weakness 1, can the author point to me which part of the theorem uses the fact that ``the other weights are updated and affect those gradients via activations”? I might be wrong, but after briefly skimming the proof, all I find is based on the assumption of the perturbation, the weights that are updated are no longer important as demonstrated in Theorem A.2.
> >
> > Regarding question 1, yes I understand the size of perturbation depends on the scale of initialization. To some extent, thm5.1 shows that if the initialization is super small, then there’s no adversarial robustness at all. But if the initialization is large, it does not mean the model is more robust. Please check the correctness of this statement.
> > The reason I said the above statement is because in thm5.1 and corollary 5.1 the size of perturbation is provided in an upper bound form. I believe it would be more appropriate to give a lower bound on the size of perturbation, demonstrating that as long as the perturbation is these large, then there’s no robustness at all.

---

> > > ### Author Response · Authors · 2023-08-11
> > > **Re: Comment**
> > >
> > > Thank you for the reply.
> > >
> > > Regarding weakness 1, there are indeed some similarities between our work and previous works on robustness for random networks. However, there are also some significant differences, and our results cannot be derived from these previous works. One important difference is that the works on random networks consider an adversarial perturbation in the direction of the gradient. In this paper (Theorem 5.1), we consider a specific perturbation which is \emph{not} in the direction of the gradient, and indeed a perturbation in the direction of the gradient would not work in our setting. The reason is that the gradient depends only on active neurons. For random networks, each neuron is active independently w.p. $1/2$, and hence it is possible to use a probabilistic argument to show such a perturbation will work. For a trained network, the active neurons depend on the data, and could possibly be chosen "adversarially" such that a perturbation in the direction of the gradient will not change the labels of the data (because the gradient changes significantly along such a perturbation). For this reason, we needed do devise a different perturbation, and the analysis is also quite different.
> > >
> > >
> > > Regarding question 1, Thm 5.1 shows that for standard initialization (i.e. which is quite large) there exists an adversarial perturbation with an upper bounded size. The reason we give an upper bound here, is to show that the perturbation is small, meaning that the network is not robust. For small initializations, we show in Section 6 an upper bound on the gradient's norm, which indicates that the network may be robust, at least to off-manifold perturbations.
> > > Specifically, in the reply "To some extent, thm5.1 shows that if the initialization is super small, then there’s no adversarial robustness at all ", it is the opposite, Thm 5.1 shows that standard initialization (i.e. large) means non-robustness.
> > >
> > > We also emphasize that in the new experiments we added, in Figure 2, the x-axis indicates the scale by which we divided the initialization of the network. This means that larger values correspond to smaller initialization scales, therefore smaller initializations resulting in larger minimal perturbations off-manifold.

---

> > > > ### Comment · Reviewer_5bBJ · 2023-08-17
> > > >
> > > > I thank the author for answering my questions.

---

### Official Review · Reviewer_ZcHg · 2023-07-07

**Soundness:** 3 good
**Presentation:** 3 good
**Contribution:** 3 good
**Rating:** 6
**Confidence:** 4

**Summary:**

The paper shows that on two-layer neural networks trained using data which lie on a low dimensional linear subspace that standard gradient methods lead to non-robust neural networks, that networks which have large gradients in directions orthogonal to the data subspace, and are susceptible to small adversarial L2-perturbations in these directions. The paper shows that decreasing the initialization scale of the training algorithm and adding L2 regularization, can make the trained network more robust to adversarial perturbations orthogonal to the data.

**Strengths:**

The paper is well written and organized, and is innovative and original. The structure of the paper is clear and rigorous.

**Weaknesses:**

The experiments are insufficient, the number of data points used to evaluate the proposed method is few.

**Questions:**

1.Does the proposed method which decreasing the gradient in directions orthogonal to the data subspace hurt the accuracy on clean data?
2.The paper claim that large gradients exists in directions orthogonal to the data subspace, and are susceptible to small adversarial L2-perturbations in these directions, do other adversarial perturbations such as L1 or L_{\infty} also exist in directions orthogonal to the data subspace?
3.Is the proposed method effective in deeper networks?
4.small typo errors,e.g. line 287,extra )

**Limitations:**

The number of data points is few, it's better to evaluate on more data points.

---

> ### Author Rebuttal · Authors · 2023-08-09
>
> We thank the reviewer for the thorough review and constructive comments.
>
> "The experiments are insufficient, the number of data points used to evaluate the proposed method is few.":
> The experiments in the paper were done mostly for visualization purposes, with low dimensions and dataset size. We added several new experiments on larger datasets, both synthetic and MNIST/CIFAR10, please see the comment above to all the reviewers and the attached PDF. These experiments both show empirically on real datasets that they approximately lie on a low dimensional subspace, and that a small initialization scale indeed increases the robustness outside of the data subspace.
>
> Regarding the questions:
>
> 1) Our methods, that is changing the initialization scale and $L_2$ regularization, may affect the accuracy on clean data.  We note that it does not necessarily hurt it; namely, it may as well improve the accuracy. Indeed, $L_2$ regularization is often used in practice for improving accuracy, regardless of its effect on adversarial robustness. Analyzing the effect of our methods on the accuracy requires additional assumptions on the data, since our results are general in the sense that we have almost no assumptions on the data, besides residing on a low dimensional subspace.
>
> 2) In this work, we focused on adversarial perturbations w.r.t the $L_2$ norm. We believe that there are also adversarial perturbations w.r.t. other norms in directions orthogonal to the data, although proving such perturbations exist requires a different analysis, which is beyond the scope of the paper.
>
> 3) Yes, our methods are effective for deeper networks, at least empirically, as demonstrated in Figure 4 in Appendix E. In the attached PDF, we describe further experiments on a low-dimensional sphere and on the MNIST dataset using deeper networks, demonstrating the same robustness effect described in the paper.

---

> > ### Comment · Reviewer_ZcHg · 2023-08-16
> >
> > Thank to the authors for their response, I have no more questions.

---

### Author Rebuttal · Authors · 2023-08-09

We thank the reviewers for the thorough reviews and constructive comments.
In the attached PDF we add two experiments, aiming at showing empirically the effects of small initialization and that real datasets approximately lie in a low dimensional subspace.

1) To show that real datasets approximately lie in a low dimensional subspace we performed PCA on MNIST and CIFAR10, and calculated the cumulative variance. For MNIST, it reached $90\%$ of the total variance by accumulating $86$ components, and $95\%$ variance by accumulating $153$ components. Similarly, CIFAR10 reaches $90\%$ of the total variance by accumulating $98$ components, and $95\%$ variance by accumulating $216$ components. Note that MNIST is a $784$-dimensional dataset, and CIFAR10 is a $3072$-dimensional dataset. This indicates that most of the information for both of these datasets indeed lies in a low-dimensional subspace.

2) We trained a $3$-layer fully-connected neural network on two datasets for different initialization scales of the first layer of the network, and used a standard projected gradient descent adversarial attack to calculate the effect of the initialization scale on the distance from the decision boundary. The datasets are:

   - MNIST projected on a $32$ dimensional subspace using PCA.
   - $500$ random samples from a $20$-dimensional sphere which lies in a $784$-dimensional space.

    The adversarial attack is either: (a) Projected on the data subspace; (b) Projected off the data subspace (i.e. the space orthogonal to it); or (c) Unconstrained attack. Each experiment was repeated $5$ times (with a newly trained networks each time), and error bars are given in the plot.

    This experiment shows the dramatic effect of changing the initialization scale on the attacks projected off the data subspace. Also, changing the initialization scale improves robustness for the unconstrained attacks, while having almost no effect for the attacks projected on the data manifold.

---

### Decision · Program_Chairs · 2023-09-21

**Decision:**

Accept (poster)

**Comment:**

The paper investigates adversarial robustness of trained 2-layer neural networks when data lies within a low-dimensional linear subspace.  It shows that standard gradient methods lead to non-robust neural networks.  Additionally, it demonstrates that reducing the initial scale of the training algorithm can enhance the trained network's resilience to adversarial perturbations. Exploring adversarial perturbations within the context of low-dimensional subspaces is quite interesting. The paper is well-written, with clear and innovative mathematical derivations.